# Rate-induced tipping in natural and human systems

Paul Ritchie[1], Hassan Alkhayuon[2], Peter Cox[1], and Sebastian Wieczorek[2]

[1]Faculty of Environment, Science and Economy, University of Exeter, North Park Road, Exeter, EX4 4QE, UK.
[2]School of Mathematical Sciences, University College Cork, Western Road, Cork T12 XF62, Ireland.

**Correspondence:** Paul Ritchie (Paul.Ritchie@exeter.ac.uk)

**Abstract.** Over the last two decades, tipping points in open systems subject to changing external conditions have become a topic of a heated scientific debate, due to the devastating consequences that they may have on natural and human systems. Tipping points are generally believed to be associated with a system bifurcation at some *critical level* of external conditions. When changing external conditions cross a *critical level*, the system undergoes an abrupt transition to an alternative, and often
less desirable, state. The main message of this paper is that the *rate of change* in external conditions is arguably of even greater relevance in the human-dominated anthropocene, but is rarely examined as a potential sole mechanism for tipping points. Thus, we address the related phenomenon of rate-induced tipping: an instability that occurs when external conditions vary faster than some *critical rate*, usually without crossing any critical levels (bifurcations). First, we explain when to expect rate-induced tipping. Then, we use three illustrating and distinctive examples of differing complexity to highlight universal and generic
properties of rate-induced tipping in a range of natural and human systems.

## 1  Introduction

In this paper, we consider tipping instabilities in nonlinear open systems (Ashwin et al., 2012). By "open" we mean systems that are influenced by changing external conditions which we refer to as *external forcings*. In a mathematical dynamic model of an open system, such external forcings are represented by *time-varying input parameters*.

Large and abrupt changes in the state of an open system may occur when the external forcing exceeds some *critical level* (Scheffer, 2010; Lenton, 2011; Kuehn, 2011). The points in time, or in the level of forcing, at which such changes occur are commonly referred to as *bifurcation-induced tipping points* (Ashwin et al., 2012). They have been identified in many domains, including ecosystems (Scheffer et al., 1993, 2001, 2009; Siteur et al., 2014; Dakos et al., 2019; Pierini and Ghil, 2021) and the human brain (Rinzel and Ermentrout, 1998; Moehlis, 2008; Screen and Simmonds, 2010; Mitry et al., 2013;
Maturana et al., 2020), and are of particular concern under anthropogenic climate change (Lenton et al., 2008; Ashwin and von der Heydt, 2020; Arias et al., 2021; Ritchie et al., 2021; Boers and Rypdal, 2021; Boulton et al., 2022). Furthermore, it has recently been recognised that critical levels can be exceeded temporarily without causing tipping (van der Bolt et al., 2018; Ritchie et al., 2019; Alkhayuon et al., 2019; O'Keeffe and Wieczorek, 2020; Siteur et al., 2016). This occurs when the time of exceedance is short compared to the inherent timescale of the system (O'Keeffe and Wieczorek, 2020; Ritchie et al., 2021;
Alkhayuon et al., 2022).

However, there is another, less obvious potential consequence of changes in external forcing. When an external forcing changes faster than some *critical rate*, rather than necessarily by a large amount, this can lead to *rate-induced tipping points* (Stocker and Schmittner, 1997; Luke and Cox, 2011; Wieczorek et al., 2011; Ashwin et al., 2012; Ritchie and Sieber, 2016; Siteur et al., 2016; Suchithra et al., 2020; Arumugam et al., 2020; Pierini and Ghil, 2021; Wieczorek et al., 2021; Longo et al., 2021; Kuehn and Longo, 2022; Kaur and Sharathi Dutta, 2022; Hill et al., 2022; Arnscheidt and Rothman, 2022). In contrast to bifurcation-induced tipping, rate-induced tipping occurs due to fast enough changes in external forcing, and usually without exceeding any critical levels by external forcing. Such tipping points are much less widely known, and yet are arguably even more relevant to contemporary issues such as climate change (Lohmann and Ditlevsen, 2021; Clarke et al., 2021; O'Sullivan et al., 2022), ecosystem collapse (Scheffer et al., 2008; Vanselow et al., 2019; van der Bolt and van Nes, 2021; Neijnens et al., 2021; Vanselow et al., 2022), and the resilience of human systems (Witthaut et al., 2021).

This paper combines a review writing style with new results to make the concept of rate-induced tipping points accessible to a wide scientific audience. Even though the phenomenon is rarely discussed by scientists and policy makers, we argue that it is ubiquitous and likely to be prevalent in many open systems. The rationale is that the current human-dominated era of Earth history, which has been called the Anthropocene (Crutzen, 2002; Crutzen and Stoermer, 2021), is characterized by systems (e.g. climate, ecosystems, infrastructures, economy) that are subject to fast-changing external conditions, and thus kept far from the changing equilibrium, by human activity. These circumstances of rapidly changing external forcing are precisely the conditions that can lead to rate-induced tipping. Rate-induced tipping is therefore especially relevant to the contemporary period, even though the phenomenon is not widely known or understood. By contrast, bifurcation-induced tipping in a system that is forced slowly towards a threshold, and thus stays close to or tracks the changing equilibrium (Ashwin et al., 2012) , is much more widely understood but is less relevant in a rapidly forced system (Ritchie et al., 2021). We demonstrate this through analysis of three distinct dynamic models of natural and human systems with differing complexity: a predator-prey ecosystem; the large-scale ocean circulation; and an electrical power grid network.

Rate-induced tipping occurs when the system deviates too much from the changing stable equilibrium and crosses some *threshold*. Here, we focus on examples of what Wieczorek et al. (2021) call a *regular threshold*. Easily verifiable criteria for the occurrence of rate-induced tipping, such as *threshold/basin instability*, are identified in all of these examples. Furthermore, we uncover universal features of rate-induced tipping. These include multiple *critical rates of change*, due to the interaction of different timescales of the external forcing with the inherent timescales of the system. Finally, we highlight important phenomena, such as *return tipping*, that are non-obvious and can be easily overlooked. For further reading on different threshold types, see O'Sullivan et al. (2022) for an example of an elusive *quasithreshold*, and Lohmann and Ditlevsen (2021) for an example of what appears to be a fractal-like *irregular threshold*.

## 2   When to expect Rate-induced tipping?

A system is known to be susceptible to rate-induced tipping if the state the system currently resides in is *threshold unstable* (O'Keeffe and Wieczorek, 2020; Wieczorek et al., 2021). Suppose that, for a given initial level of external forcing, the

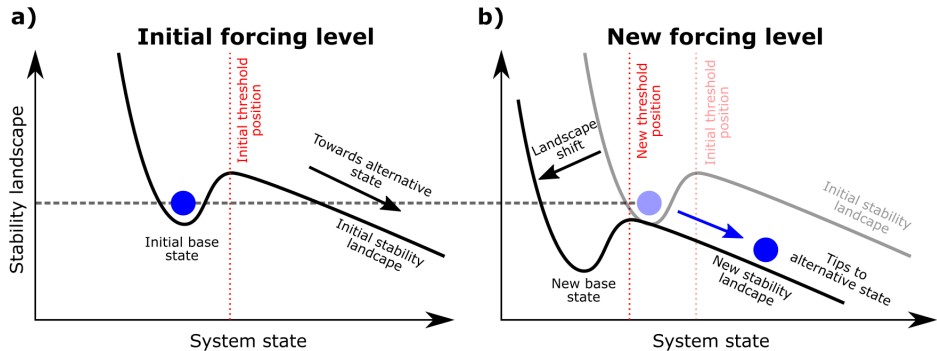

**Figure 1. Schematic illustration of threshold instability**. (a) The stability landscape of the system at an initial level of forcing. The well represents the base state, the hill top defines the threshold (indicated by the red dashed vertical line), and the (blue) ball indicates the current state of the system. To the left of the hill top (threshold), the ball rolls into the well, meaning the system converges to the base state. To the right of the hill top, the ball runs away, indicating tipping to an alternative state. (b) The stability landscape at a new forcing level. Note that the initial base state is on the other side of the new hill top (threshold). Therefore, if the system is at the initial base state and the forcing switches sufficiently fast to the new level, the ball will run away, meaning the system will tip to an alternative state.

system resides in a stable equilibrium (though in general it can be a stable limit cycle or an even more complicated attractor). This equilibrium will be referred to as the *base state*. One way of depicting threshold instability is with a moving stability landscape, as illustrated by Figure 1, where the base state is represented by the blue ball in the well in panel (a).[1] The hill top defines the position of the *threshold* (indicated by the vertical red dashed line). If the ball is to the left of the hill top (threshold), it will roll into the well and the system will converge to the base state. Whereas, if the ball is to the right of the hill top, it will roll in the opposite direction and the system will tip to some *alternative state*. The alternative state may be a *different stable state* for a multistable system (Scheffer et al., 2008; O'Keeffe and Wieczorek, 2020; Halekotte and Feudel, 2020; Lohmann et al., 2021; Slyman and Jones, 2023), or a *transient state* for an excitable (possibly monostable) system (Wieczorek et al., 2011; Vanselow et al., 2019; O'Sullivan et al., 2022).

In contrast to bifurcation-induced tipping, the change in the forcing usually does not cause any qualitative change in the stability landscape but instead shifts its position. If the threshold moves past the initial position of the base state for a new forcing level, as shown in panel (b), the base state is said to be *threshold unstable* on varying the forcing (Wieczorek et al., 2021). In the case when the threshold is a basin boundary of two attractors in a multistable system, the system is said to be *basin unstable* (O'Keeffe and Wieczorek, 2020). The threshold/basin instability condition gives the forcing shift magnitude that enables rate-induced tipping. In general, one can prove that threshold/basin instability is sufficient for the occurrence of rate-induced tipping: There is an external forcing that gives rate-induced tipping if the system is threshold/basin unstable (Kiers and Jones, 2020; Wieczorek et al., 2021). In many examples, including those considered here, we find that threshold instability

---

[1]We note that this example is for illustrative purposes. In general, the base state can be non-stationary, the system may reside near rather than in the base state, and not all dynamical systems can be characterised by a stability landscape; see for example Zhou et al. (2012).

appears to be both necessary and sufficient for the occurrence of rate-induced tipping: There is an external forcing that gives rate-induced tipping if and only if the system is threshold/basin unstable.

The rigorous result can be understood intuitively as follows. Consider a change in the level of the forcing that gives threshold/basin instability as depicted in Figure 1. If the forcing changes from the initial level to the new level at a sufficiently slow rate, the ball remains in the well and the system is said to *track* the moving base state. If the forcing switches at a sufficiently fast rate, the initial ball finds itself on the other side of the hill top (threshold) and *tips* to an alternative state. Thus, there will be at least one intermediate *critical rate of change* at which there is a transition between tracking and tipping. Once it is known that a system is threshold/basin unstable, and thus susceptible to rate-induced tipping, the goal is to find the critical rate, or even multiple critical rates, for a given profile (shape) of external forcing. In the next section, we give a more precise description of critical rates.

## 3   Defining critical rates

Let's denote the time-varying external forcing with $\lambda$. The *level* of the forcing at a time $t$ is simply the value of $\lambda$ at this time $t$. However, defining *critical rates* of change in external forcing is more subtle. On the one hand, different external forcings will have different physical units and be different, often nonlinear functions of time. On the other hand, we would like to quantify critical rates of change in a uniform way, that is independent of the physical units and the *temporal profile* of the forcing. Therefore, we introduce a *rate parameter* $r$ in units inverse second (or day, year, etc.), write the external forcing as $\lambda(rt)$, where $u = rt$ is dimensionless, and work with $r$ as the main input parameter. Most importantly, we define a *critical rate* as a special value of $r$ at which rate-induced tipping occurs, while the shift magnitude of $\lambda(rt)$ remains fixed.

To avoid confusion between the rate parameter $r$ and the rate of change of external forcing $d\lambda/dt$, we note that

$$d\lambda/dt = \frac{d\lambda}{du}\frac{du}{dt} = r\,\frac{d\lambda(u)}{du},$$

has units of $\lambda$ per second, depends on $r$ as well as on the profile of $\lambda(u)$, and may itself be a function of time. In other words, the rate parameter $r$ quantifies the rate of change of external forcing with a given profile. Furthermore, if the forcing $\lambda$ itself is a physical rate of some sort (e.g. freshwater flux into the North Atlantic, measured in Sverdrups – millions of cubic metres per second, or population growth rate measured in individuals per unit area per year, from examples in Section 5), $\lambda(rt)$ will be the level of this rate at time $t$, referred to as the level of the forcing, and $r$ will quantify the rate of change of this rate, referred to as the rate of change of the forcing.

## 4   Rate-induced tipping in a simple model

In natural and human systems, tipping points are often associated with crossing a critical level of the forcing, defined by a dangerous (e.g. fold) bifurcation for the frozen system with fixed-in-time forcing, causing a catastrophic, abrupt and irreversible change to the state of the system (Thompson et al., 1994; Thompson and Sieber, 2011). This type of tipping is commonly referred to as bifurcation-induced tipping (or B-tipping) and is illustrated by Figure 2(a), (d) (Ashwin et al., 2012). Suppose the

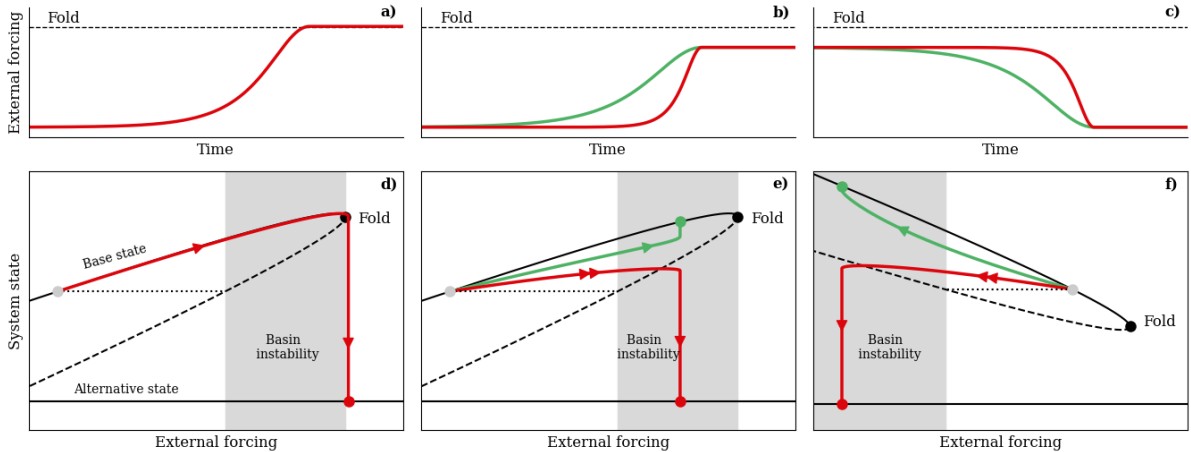

**Figure 2. Illustration of bifurcation-induced, rate-induced, and return tipping.** (a) – (c) Time profiles of ramp external forcing, (d) – (f) corresponding system response to these forcing profiles. (a), (d) Bifurcation-induced tipping: slow change in external forcing past a critical level (Fold bifurcation) causes tipping. The tipping occurs for any rate of change of the forcing past the Fold. (b), (e) Rate-induced tipping: the system fails to adapt to a too fast change in external forcing (red) even though the forcing never crosses the critical level. For a slow enough change of the forcing the system tracks the moving base state and avoids tipping (green). (c), (f) Return tipping: avoiding bifurcation-induced tipping by reversing the trend in the external forcing too quickly can lead to rate-induced tipping on decrease of the forcing. Branches of stable equilibria are denoted by black solid curves, and branches of unstable equilibria are denoted by black dashed curves. Stable and unstable branches meet at a Fold bifurcation (black dot). The system starting from the grey dot is basin unstable for forcing shift magnitudes that end in the grey region of basin instability.

system starts from the base state on the upper branch of stable equilibria for the frozen system, as indicated by the grey dot in Figure 2(d). Initially, as external forcing changes slowly (Figure 2(a)), the state of the system (the red trajectory in Figure 2(d)) tracks the moving base state (the branch of stable equilibria). However, once external forcing reaches the fold that defines the critical level of the forcing, the base state disappears (the branch of stable equilibria terminates), and the system subsequently undergoes a catastrophic transition to the alternative stable state. Crucially, the tipping occurs for any rate of change of the forcing past the Fold. The alternative state is often a less desired state, such as an extinction state in an ecosystem (O'Keeffe and Wieczorek, 2020), collapse of an ocean circulation (Alkhayuon et al., 2019), or a blackout on a power grid network (Budd and Wilson, 2002). However, it could also be a more desired state, such as a well-being state for developing countries (Mirza et al., 2019).

Figure 2 introduces a subtle but crucial difference to previous examples that have considered B-tipping (Lenton et al., 2008; Scheffer et al., 2009; Ritchie et al., 2021) and that is to apply a tilt[2] to the bifurcation structure (O'Keeffe and Wieczorek, 2020,

---

[2]Many conceptual models of bifurcation-induced tipping use the one-dimensional normal form of a fold (saddle-node) bifurcation to illustrate and study the phenomenon. While all systems that exhibit a fold bifurcation are topologically equivalent to its normal form sufficiently close to the bifurcation point, the behaviour of the branches of equilibria will typically be different away from the bifurcation point. Our tilted branches incorporate simple deviations from the normal-form behaviour expected in higher-dimensional systems away from the bifurcation point.

Sec.7), see Methods for further details. This important distinction introduces the possibility of a different form of tipping known as rate-induced tipping (or R-tipping). Unlike B-tipping in Figure 2(a), (d), where crossing a critical level of the external forcing causes a catastrophic transition, R-tipping occurs when the system fails to adapt to a too rapidly changing external forcing, usually without crossing any critical levels. Figure 2(b), (e) considers two scenarios where the change in the level of external forcing is the same, but occurs at different rates. Most importantly, the forcing stops in the (grey) region of basin instability, and never crosses the critical level defined by the Fold bifurcation point. For a slow rate of change in external forcing (green trajectory), the system is able to continually adapt to the moving base state and tracks the stable branch of equilibria without tipping. However, for a slightly faster rate of change in external forcing (red trajectory), the system is unable to adapt to the moving base state and undergoes R-tipping to the alternative stable state.

If a system is thought to be approaching a B-tipping event, then a natural option would be to reverse the external forcing to avoid crossing a largely unknown critical level. However, fast reversals in the forcing could introduce a new problem that has been largely overlooked, namely *return tipping* (O'Keeffe and Wieczorek, 2020). Figure 2(c), (f) illustrates such a scenario for a Fold bifurcation structure tilted down. Suppose that external forcing has caused a system to approach close to the Fold bifurcation. Reversing the forcing slowly will allow the system to closely track the moving base state (the branch of stable equilibria) as shown by the green trajectory. However, a too fast reversal may give rise to R-tipping on return if the system is basin unstable on reversal of the forcing. Then, the end result is opposite to what was intended. Although B-tipping is avoided, the system, rather surprisingly, R-tips to the alternative stable state (red trajectory). Therefore, in general, reversing external forcing as quickly as possible does not guarantee avoiding tipping.

Figure 3 provides a more in-depth analysis of the tilted saddle-node model considered in Figure 2(a), (b). Let us assume initially that the external forcing has the profile of a *nonlinear shift ramp*; see Methods for details. Three sample time series of shift forcing profiles between the same levels but at different rates are given by a concatenation of the left half of a colour curve and the black dashed curve in Figure 3(a); these are similar to external forcings used in Figure 2(a) and (b). The corresponding response of the system is depicted in Figure 3(b). For the slowest change in external forcing (the blue dashed trajectory) the system is able to adapt to and track the changing base state. However, if the rate of change in external forcing becomes too fast (the orange and purple dashed trajectories), then the system fails to adapt to the changing base state and R-tips to the alternative state.

The critical rate, which determines the onset of R-tipping, will depend on how much the external forcing is changed by. The black dashed curve in Fig. 3(c) shows the critical boundary for the ramp external forcing, separating regions of tipping (coloured) from no tipping (white), in the plane of the rate parameter against the change in the forcing level, referred to as the peak change. B-tipping occurs if the external forcing crosses the Fold bifurcation without returning. Indeed, for very small rate parameters (slow rates) the critical boundary asymptotes to the distance required to reach the Fold (indicated by the thin black line). However, for larger rate parameters (faster rates), tipping can occur before the Fold is transgressed because of R-tipping. The three coloured dots correspond to the external forcing parameters used in Figure 3(a) and (b). Notice that the blue dot is in the white region, signifying tracking, whereas the other two dots are within the coloured regions denoting tipping for the ramp forcing. For very large rate parameters, the critical boundary asymptotes to the basin instability boundary: the smallest

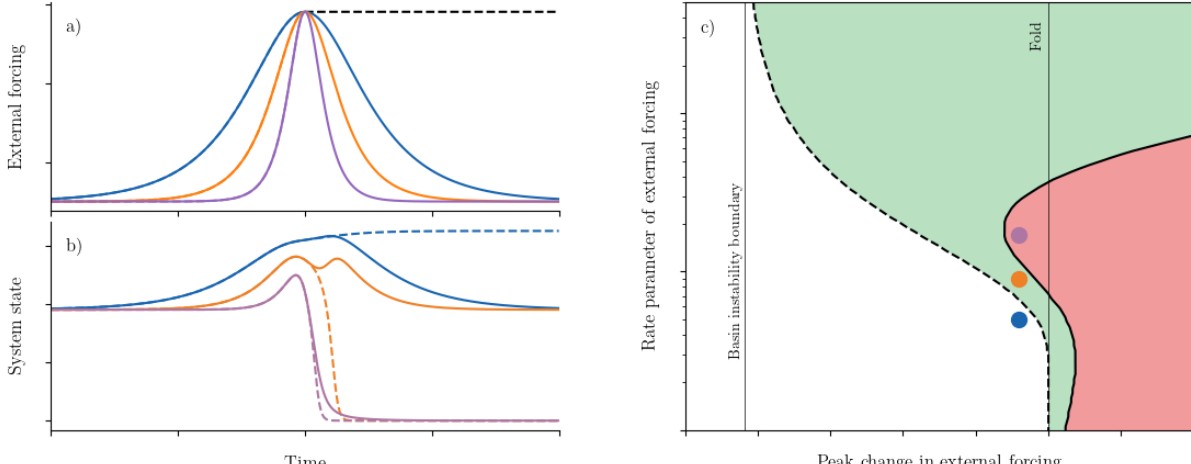

**Figure 3. Tipping for a nonlinear shift ramp and return profiles in the tilted fold example.** (a) Time profiles of ramp and return (impulse) external forcings, and (b) system response to these forcing profiles. Forcing profiles vary between the same minimum and maximum levels but at different rates: slow (blue), medium (orange) and fast (purple). Ramp forcing profiles in (a) are given by a concatenation of the left half of a colour curve and the black dashed curve. The corresponding system responses in (b) are given by a concatenation of the left half of the solid curve and the dashed curve of the same colour. Return forcing profiles in (a) and the corresponding system responses in (b) are given by the solid colour curves. (c) Tipping diagram for ramp and return profiles; note the logarithmic scale for the rate parameter. Critical boundaries separate regions of tracking from tipping for the ramp forcing profile (black dashed curve) and for the return forcing profile (solid black curve). White region: tracking for ramp and return profiles; green region: tipping for ramp profile, tracking for return profile; red region: tipping for ramp and return profiles.

change in the level of external forcing that gives basin instability as described in Figure 1. In summary, the ramp forcing with a peak change past the basin instability boundary and below the Fold level gives rise to R-tipping with a single critical rate. This critical rate decreases with the peak change.

The dynamics become more interesting when the external forcing is reversed back to its initial level after reaching its peak
level. To illustrate how, we will now consider external forcing that has a profile of a *symmetric impulse*, referred to as a return forcing profile;[3] see Methods for further details. Three sample time series of return forcing profiles with the same peak change but different rates are given by the colour curves in Figure 3(a). The critical boundary for the return external forcing is given by the solid black curve in Figure 3(c), which separates regions of tracking (green and white) from tipping (red), and is very different from the dashed curve for the ramp external forcing. To be more specific, the green (points of return) region
corresponds to scenarios where tipping is prevented by reversing the external forcing. The red (points of no return) region corresponds to scenarios where tipping still occurs despite reversing the external forcing. Previously, it has been shown for B-tipping that safely overshooting a critical level by a given distance can be achieved, provided the reversal in external forcing

---

[3]As a generalisation of a symmetric impulse, one could consider a pulse where the increase towards the peak level and the decrease back to the initial level occur at different rates.

is faster than some critical rate. However, the added possibility of R-tipping owing to the tilted bifurcation structure, combined with the symmetric return forcing (see Methods for further details), means that *multiple critical rates* can arise for a fixed peak change in the return forcing profiles; this is illustrated by the S-shaped solid black curve in Figure 3(c). In a symmetric return forcing, multiple critical rates emerge because there is competition between sufficiently slow approach towards the Fold required to avoid R-tipping, and sufficiently fast reversal required for safe overshoots of the Fold.

In the green region to the left of the vertical Fold line, reversing the forcing prevents the system from an impending R-tipping that would occur if the forcing were not to be reversed. An example is given by the solid orange curve in Figure 3(b). Interestingly, there is also a small red region to the left of the vertical Fold line. This region gives rise to two critical rates for a fixed peak change, which bound a (red) sub-interval of the rate parameter where R-tipping is not prevented by return forcing. An example is given by the solid purple curve in Figure 3(b).

For small overshoots of the Fold, even greater complexity is possible with the potential of three critical rates and two (red) tipping sub-intervals for a fixed peak change of return forcing. For very small rate parameters (the red region to the right of the lower part of the vertical Fold line), B-tipping occurs and cannot be prevented by reversing the forcing. However, for the same peak change and larger rate parameters (the slightly wider green region to the right of the lower part of the vertical Fold line), it becomes possible to prevent B-tipping and avoid R-tipping upon return. Keeping the peak change fixed and increasing the rate parameter even more (the red region to the right of the middle part of the vertical Fold line) prevents B-tipping but triggers R-tipping, meaning that the system tips again despite reversing the forcing. Then, for the same peak change and very large rate parameters (the green region to the right of the upper part of the vertical Fold line), both B-tipping and R-tipping upon return can be prevented again, but for a different reason - the system processes are too slow to react to a fast forcing impulse.

## 5 Rate-induced tipping in ecology and climate

We now consider an example from ecology, namely that of a predator-prey system, which models the time evolution of plant and herbivore biomass densities (Scheffer et al., 2008). The model has been proposed to conceptually study tipping points in bistable ecosystems with a non-monotone functional response. Examples of such systems can be: the dominance shift between submerged macrophytes and phytoplankton (Scheffer et al., 1993), coral reefs and macro-algae (Hughes, 1994), or the transition of kelp forests into sea urchin barrens that are dominated by crustose coralline algae (Steneck et al., 2002).

In this example, changes in environmental conditions affect the plant growth rate and herbivore mortality rate simultaneously, and play the role of external forcing. Such a scenario can be considered possible under climate change, where plants benefit from the fertilisation effect due to increasing levels of $CO_2$ (Reich et al., 2014), but the resulting increased temperatures are detrimental to herbivores (Lacetera, 2019). The plant growth rate and herbivore mortality rate vary within a range where the ecosystem has two stable equilibria. The stable coexistence equilibrium is the base state. The stable plant-only equilibrium with no herbivores is the alternative stable state.

We consider three sample time profiles of a nonlinear shift ramp external forcing, shown in Figure 4(a), with the same change in the level of environmental conditions but at different rates. The resulting impacts on the herbivore biomass are

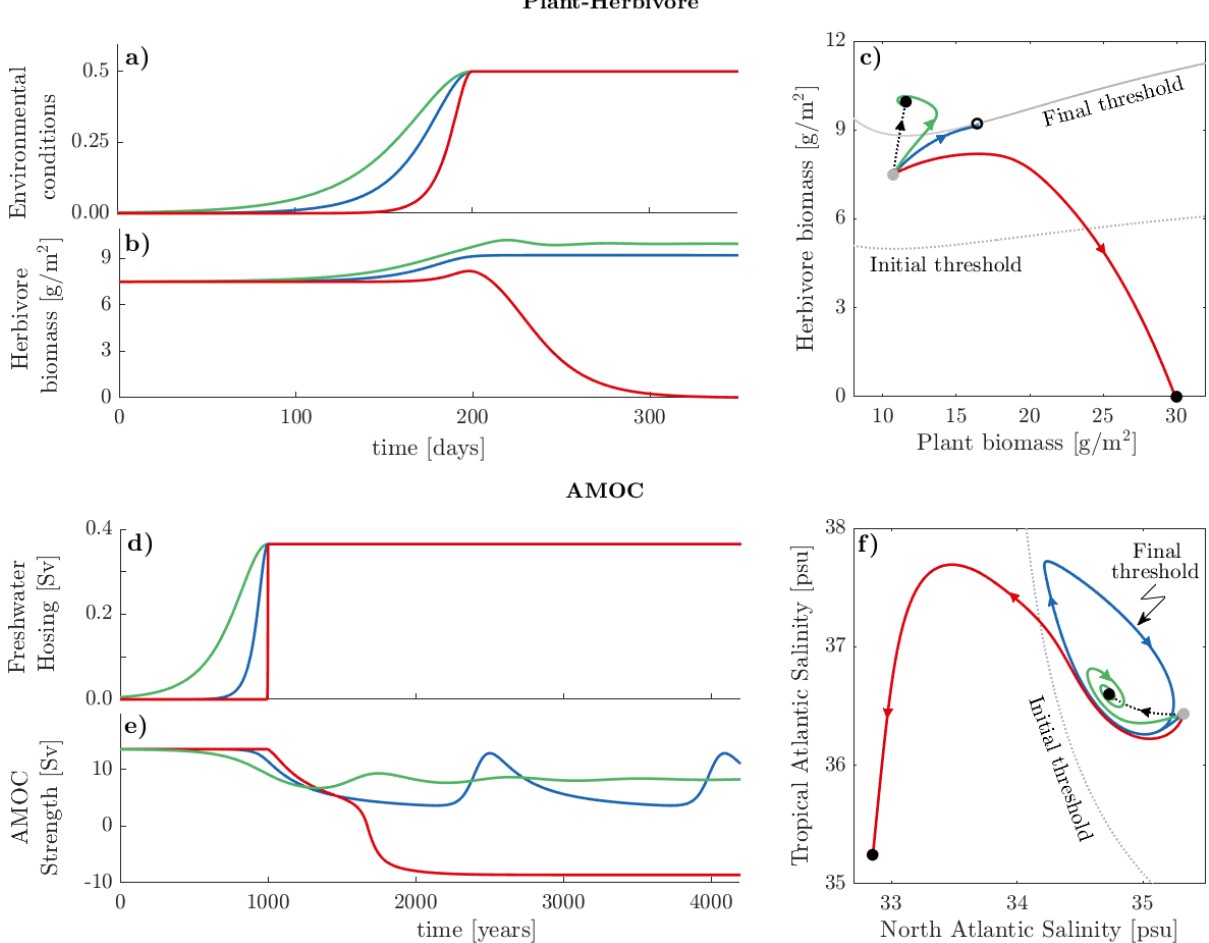

**Figure 4. Rate-induced tipping in the plant-herbivore (top row) and AMOC (bottom row) models.** Time series of the external ramp forcing profiles: (a) dimensionless environmental conditions and (d) freshwater hosing; see Methods for further details. Time series of the system responses to the changing (b) environmental conditions and (e) freshwater hosing, for three different rates of change (different colours). The system responses to external ramp forcing profiles in the phase plane of (c) the plant and herbivore biomass, and (f) the North and Tropical Atlantic salinities. See Supplementary Video 1 and Supplementary Video 2 for animated versions of (a)–(c) and (d)–(f) respectively.

shown in Figure 4(b). For the slowest (green) change in environmental conditions, the ecosystem tracks the moving base state. Herbivores increase slightly, which represents the continued presence of coexisting plants and herbivores. Contrast this to the fastest (red) change in environmental conditions, which causes R-tipping to the alternative stable state. In this scenario, herbivore population declines to zero and the ecosystem becomes entirely dominated by plants.

Figure 4(c) illustrates the underlying dynamics in the phase plane of the plant and herbivore biomass. For the initial level of environmental conditions, the base state is indicated by the grey dot. The shift in the level of environmental conditions changes the position of the base state, and this change is indicated by the dotted black line. The black dot at the other end of the dotted line indicates the base state for the final level of environmental conditions. Further ramifications of the shift in environmental conditions include changes in the basin of attraction of the base state. The basin of attraction shifts (from the dotted curve to the solid grey curve) such that the initial base state (the grey dot) is not contained in the basin of attraction of the final base state (the region above the solid grey curve). A consequence of basin instability is that the behaviour of solutions starting near the initial base state depends on the rate of change of environmental conditions as follows. For slow rates, the system is able to continually adapt and remain within the changing basin of attraction of the base state so that solutions converge to the final base state (the green trajectory). However, for a sufficiently fast rate, the ecosystem fails to track the fast moving base state and R-tips to the final alternative stable state indicated by the second black dot (the red trajectory). This rate-sensitive behaviour is expected due to basin instability. What may be surprising is that, at a critical rate of change in environmental conditions, the corresponding solution converges to an unstable *edge state* (black circle) (Wieczorek et al., 2021) on the basin boundary of the final base state (the blue trajectory).

Next, we consider an example from climate, specifically the possible collapse of the Atlantic Meridional Overturning Circulation (AMOC) under global warming. The AMOC forms part of the global thermohaline circulation, which is a large-scale ocean circulation current driven by temperature and salinity gradients. The AMOC contributes to the relatively mild climate in Western Europe by transporting heat from the Tropics to the North Atlantic. Once the warm salty waters reach the North Atlantic, they cool down and become denser. The higher density of these waters causes sinking (or overturning), followed by a return to the Tropics along the bottom of the ocean. However, the AMOC can be easily disturbed by contemporary climate change. The amount of freshwater added to the North Atlantic (referred to as freshwater hosing) may increase under climate change, for example due to the melting of the Greenland Ice Sheet, changes in precipitation patterns, or both. Specifically, Stocker and Schmittner (1997) and Lohmann and Ditlevsen (2021) show in coupled climate models that the AMOC can collapse under sufficiently fast rates of change in either $CO_2$ emissions or freshwater hosing. Additionally, R-tipping of the AMOC has been observed in a global oceanic box model (Alkhayuon et al., 2019).

Here, we work with the global oceanic box model for the AMOC (Wood et al., 2019), in which changing freshwater hosing plays the role of external forcing. The forcing varies within a range where the AMOC model has two stable equilibria. The stable AMOC-On equilibrium is the base state. The stable AMOC-Off equilibrium is the alternative stable state.

Figure 4(d) shows three sample time profiles of ramp forcing that increase the freshwater hosing from zero to the same non-zero level, but each at a different rate. The response of the AMOC to these freshwater forcing scenarios is shown in Figure 4(e). For the slowest (green) change in freshwater hosing, there is tracking of the moving base state. The AMOC strength suffers a slight drop, but ultimately remains in the AMOC-On state. However, for the fast (red) change in freshwater hosing, there is R-tipping to the alternative stable state. The AMOC strength declines to a point of complete collapse.

The phase portrait in the plane of the salinities in the North and Tropical Atlantic boxes in Figure 4(f) illustrates the underlying dynamics. The grey dot shows the base state for the initial level of freshwater hosing. Increasing the freshwater hosing

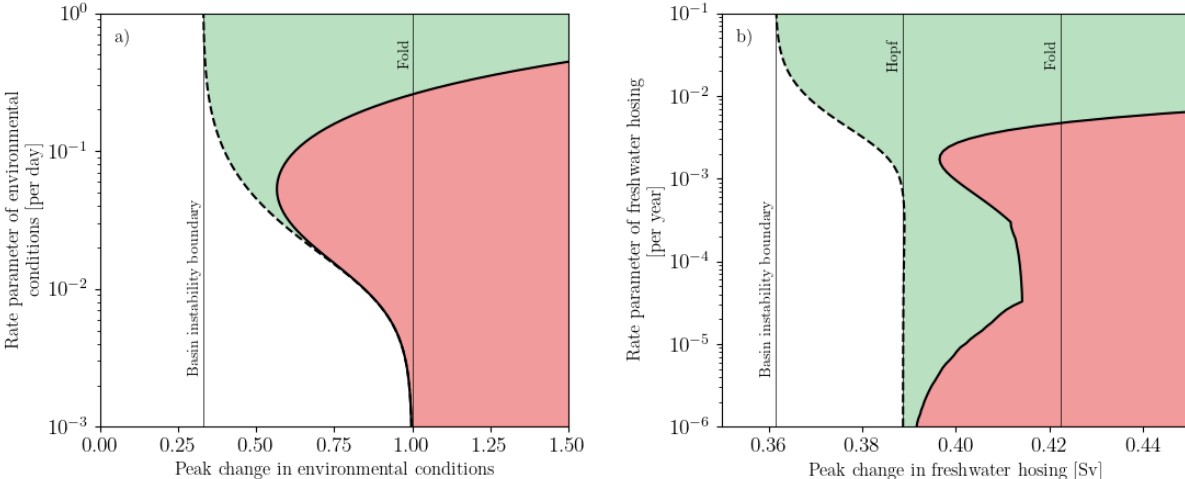

**Figure 5. Tipping diagrams for (a) the Plant-Herbivore and (b) AMOC models.** Critical boundaries separate regions of tracking from tipping for the ramp forcing profile (black dashed curve) and for the return forcing profile (solid black curve). White region: tracking for ramp and return forcing profiles; green region: tipping for ramp forcing profile, tracking for return forcing profile; red region: tipping for ramp and return forcing profiles.

shifts the base state as indicated by the black dotted line. The black dot at the other end of the dotted line indicates the base state at the final level of freshwater hosing. Notice that the basin of attraction of the final base state (the region enclosed by the blue periodic orbit) does not contain the initial base state (the grey dot). Therefore, the initial base state is basin unstable, and R-tipping from the base state to the alternative stable state (lower black dot) will occur for sufficiently fast shifts in the level of freshwater hosing. For slow rates of increase in freshwater hosing, the system continually adapts and remains within

the changing basin of attraction of the base state so that solutions converge to the final base state (the green trajectory). On the other hand, for sufficiently fast rates of increase in freshwater hosing, the system is unable to adapt and falls outside the changing basin of attraction of the base state, causing solutions to converge to the final alternative state (the red trajectory). At the critical rate of increase in freshwater hosing, surprising behaviour is again observed as the corresponding solution converges to a repelling periodic orbit defining the basin boundary of the final base state (the blue trajectory).

To validate our choice of the simple tilted saddle-node model in Section 4, we plot in Figure 5 the tipping diagrams for the Plant-Herbivore and AMOC models, showing regions of tracking (white), points of return (green) and points of no return (red). The tipping diagrams for the Plant-Herbivore model in Figure 5(a) and the AMOC model in Figure 5(b) are very similar to the tipping diagram for the simple model in Figure 3(b), although there are some differences. In both examples, there is a region of basin instability that gives rise to R-tipping for shift forcing profiles with a peak change that does not cross any critical levels.

This means that the simple model with a tilted bifurcation structure indeed captures non-obvious tipping phenomena found in higher-dimensional systems. In Figure 5(a), the (red) region of points of no return extends to the left of the critical level (the black vertical Fold line), but there are only up to two critical rates for the return forcing profile with a fixed peak change, and

the (green) region of points of return vanishes for small rate parameters. In Figure 5(b), there are up to three critical rates for the return forcing profile with a fixed peak change, but the (red) region of points of no return does not extend to the left of the critical level (the black vertical Hopf line). Additionally the (black) boundary has small smooth 'wiggles' that appear as non-smooth corners. A similar 'wiggling effect' near a Hopf bifurcation has been observed in O'Keeffe and Wieczorek (2020).

## 6   Rate-induced tipping in power grid networks

R-tipping instabilities are not confined to natural systems, but can occur in any system, including human systems. One example is the energy sector and power grid networks (Suchithra et al., 2020). Crucially, electricity needs to be used as soon as it is produced since it cannot be stored easily (Mokrian et al., 2006). Therefore, providing a near constant voltage of electricity to millions of homes, while power demand varies seasonally, daily, and may spike during major events, is a technological challenge (Mahmud and Zahedi, 2016). Some noticeable examples of power blackouts or near misses are: the Northeast USA blackout in 2003 caused by a series of faults in local control systems (Hu et al., 2016); and the near miss blackout in England following the conclusion of the Euro 1990 semi-final (Swarup, 2007).

The latter example in particular was arguably the result of R-tipping effects. The power demand on the network, following the conclusion of the football match, was expected to be high. Hence, the national grid took measures to ensure the network would be able to cope with the high power demand. However, the national grid failed to envisage the match going to extra time and penalties. Thus, the rapid increase in power demand, following the eventual conclusion of the match, gave controllers insufficient time to react. Therefore in this case, the limiting factor was not the peak in demand, but instead the rate at which the demand on the network rose.

Here, we use a conceptual model of a power grid network (Dobson et al., 1988; Dobson and Chiang, 1989; Budd and Wilson, 2002), where changing power demand plays the role of external forcing. The situation with stable states in this model is more complicated than in the previous examples. The power demand changes within a range where there are infinitely many stable equilibria with the same fixed voltage magnitude and a $2\pi$ difference in the phase angle. Since each of these stable equilibria has the same voltage magnitude, they can be thought of as a single base state of the system. Furthermore, there are two alternative states. An *alternative transient state* is a temporary drop in the voltage magnitude accompanied by a $2\pi$ shift in the phase angle, caused by the coupling within the system. This state corresponds to a transition between two neighbouring stable equilibria within the base state. An *alternative stable state* is at zero voltage magnitude, and corresponds to electrical blackout.[4]

We now demonstrate that a rapid increase in the level of power demand can lead to R-tipping in the form of different disruptions in power supply. First, we note that for slow enough increases in the level of power demand, the power grid network always tracks the moving base state (not shown here). Then, in Figure 6, we show the response of the power grid network to ramp and return forcing profiles with higher rates of change.

---

[4]In the model, the voltage drops to negative infinity, but we restrict to physically relevant non-negative voltage magnitudes.

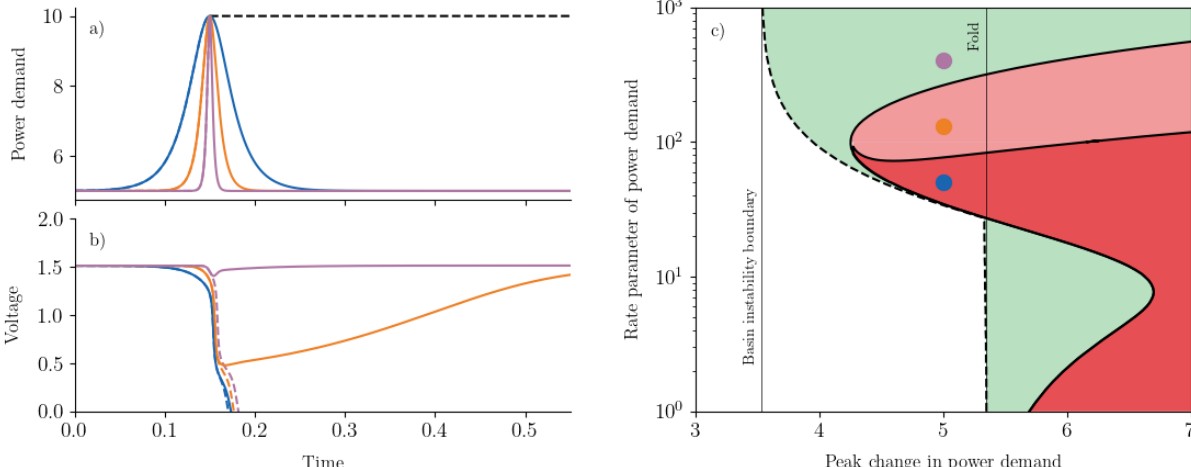

**Figure 6. Tipping for ramp and return forcing profiles in the power grid model.** (a) Time profiles of ramp and return (impulse) power demand, and (b) the resulting voltage response for these power demand (forcing) profiles. Power demand profiles vary between the same minimum and maximum levels but at different rates: slow (blue), medium (orange) and fast (purple). Ramp forcing profiles are given by a concatenation of the left half of a colour curve and the black dashed curve. (b) The corresponding system responses given by a concatenation of the left half of the solid curve and the dashed curve of the same colour. Return forcing profiles in (a) and the corresponding system responses in (b) are given by the solid colour curves. (c) Tipping diagram of ramp and return forcing profiles. Critical boundaries separate regions of tracking from tipping (blackout) for both, the ramp profile (black dashed curve) and for the return profile (black solid curve). For return forcing profiles, reversible tipping is also possible through a phase slip. White region: tracking for ramp and return profiles; green region: irreversible tipping to blackout for ramp profiles, tracking for return profiles; light red region: irreversible tipping to blackout for ramp profiles, reversible tipping to phase slips for return profiles; dark red region: irreversible tipping to blackout for ramp and return profiles.

We start with three ramp shifts that vary between the same levels of power demand, without crossing any critical levels, but at different rates. These are given by a concatenation of the left half of a colour curve and the black dashed curve in Figure 6(a). Owing to the higher rates of change, all three ramp shifts cause R-tipping to the alternative stable state, resulting in blackout; see the dashed curves in Figure 6(b). However, reversing the power demand (see solid colour pulses in Figure 6(a)) can avoid blackout and restore the power grid network to the base state if the reversal is fast enough (see the corresponding solid colour responses in Figure 6(b)). The slowest (blue) reversal is too slow to avoid *irreversible R-tipping* to blackout. The medium-rate (orange) reversal avoids blackout, but involves *reversible R-tipping* to the alternative transient state - a temporary voltage drop followed by a long recovery towards the base state. The fastest (purple) reversal avoids both types of R-tipping and does not cause any noticeable disruptions to power supply.

The tipping diagram for ramp and return forcing profiles in Figure 6(c) has, in addition to multiple critical rates already observed in the previous examples, an important new feature. Owing to the existence of two alternative states, there are two different red regions corresponding to different types of R-tipping for a return forcing profile: irreversible R-tipping to blackout (dark red region) where the voltage magnitude drops to zero permanently, and reversible R-tipping (light red region) involving a

temporary drop in the voltage magnitude accompanied by a $2\pi$ phase slip. The three scenarios illustrated in Figure 6(a) and (b) correspond to the three coloured dots in the tipping diagram in Figure 6(c). Note that all three dots are located past the basin instability boundary and below the critical level of power demand (the black vertical Fold line), meaning that these transitions are purely rate-induced.

## 7 Conclusions

We have shown that many natural and human systems can experience rate-induced tipping (R-tipping). Such instabilities usually occur for sufficiently fast increases in external forcing, and despite never crossing any critical levels of external forcing. In other words, systems are able to continually adapt to a moving base state and therefore avoid tipping when external forcing changes sufficiently slowly, but fail to adapt to or track a moving base state when external forcing changes faster than some critical rate. Reversing the external forcing can prevent a system from suffering R-tipping but the rates required for this add an additional layer of complexity in the presence of bifurcation-induced tipping (B-tipping). Previously, it has been shown that safe overshoots of critical levels for B-tipping require fast rates of change (Ritchie et al., 2021). However, faster rates of change make a system more susceptible to R-tipping. To make the concept of R-tipping accessible to a wide scientific audience, we:

- Described an easily verifiable criterion of threshold/basin instability for R-tipping to occur.

- Demonstrated basin instability and ensuing R-tipping in conceptual models of natural and human systems, including irreversible and reversible R-tipping.

- Highlighted interesting phenomena, such as multiple critical rates and return tipping, that can arise from an interplay between R-tipping and B-tipping for non-monotone forcing profiles.

The dynamic models used in this study, for representing a predator-prey ecosystem, the Atlantic overturning circulation and an electrical power grid network, are relatively simple. Since our forcing profiles are very idealised by design, we focus on the qualitative behaviour that can arise for different rates of forcing rather than on quantitative predictions. For quantitative predictions, further research on R-tipping is required in more-realistic higher-complexity models, such as state-of-the-art global circulation models, and with more-realistic forcing profiles. The base state in such models may not necessarily be a steady state (an equilibrium), but for example could take the form of a periodic orbit or even a chaotic state. This could lead to more complex tipping behaviour, such as phase tipping (P-tipping) (Alkhayuon and Ashwin, 2018; Kaszás et al., 2019; Alkhayuon et al., 2021; Ashwin and Newman, 2021; Alkhayuon et al., 2022).

R-tipping is likely to be prevalent in many systems given contemporary rates of change such as unprecedented anthropogenic climate change. This paper highlights the importance of considering how fast external forcing is changing as opposed to solely focusing on levels of change. Consequently, the actions taken to control the rate of change in forcing are equally as important as the actions taken to control the level at which forcing is halted.

### Methods

### Forcing profiles

In our analysis we consider two types of dimensionless nonlinear external forcing profiles, denoted $a(rt)$. The first is a *ramp* forcing profile, that starts close to 0, and subsequently increases continuously until reaching a peak level of 1, at time $T$. This forcing profile subsequently remains at 1.

$$a(rt) = \begin{cases} \text{sech}(r(t-T)), & 0 \leq t \leq T, \\ 1, & t > T. \end{cases} \tag{1}$$

For R-tipping scenarios with a fixed peak change, the rate of change in the forcing, quantified by the rate parameter $r$, determines if tipping occurs.

The second forcing profile we consider is a *return* (impulse) forcing, primarily used to examine the possibility of avoiding tipping. Equation (1) is modified by removing the piecewise element of the forcing for $t > T$, such that the forcing returns back to its initial level (at a mirrored rate of the approach) after reaching the peak level of 1, at time $T$:

$$a(rt) = \text{sech}(r(t-T)). \tag{2}$$

One example of a return forcing profile, given by Equation (2), is an idealised scenario to reverse the impact of anthropogenic climate change back to initial levels, i.e. via the development of technologies to remove $CO_2$ from the atmosphere (Huntingford et al., 2017).

### Conceptual model

We use a conceptual model to illustrate some of the universal features associated with R-tipping alone, and with an interplay between R-tipping and B-tipping. Using a modified (tilted) version of the normal form for a saddle-node bifurcation, a state variable, $x$, is modelled by the following single ordinary differential equation:

$$\frac{\mathrm{d}x}{\mathrm{d}t} = -x\left[(x - A - s\lambda(rt))^2 + \lambda(rt)\right], \tag{3}$$

where $s$ is a tilt parameter, $A$ provides the distance between the fold point and the alternative state, and the external forcing is given by

$$\lambda(rt) = \lambda_- + \Delta_\lambda a(rt),$$

where $a(rt)$ takes the form of a ramp or a return profile defined above. Figures 2 (a), (d), are obtained using parameter values: $T = 500$, $s = 4$, $A = 3.2$, $\lambda_- = -0.5$, $\Delta_\lambda = 0.505$, $r = 0.05$ and initial condition at the location of the base state for

$\lambda = \lambda_-$. Figures 2 (b), (e) are obtained using parameter values: $T = 500$, $s = 4$, $A = 3.2$, $\lambda_- = -0.5$, $\Delta_\lambda = 0.4$, $r = 1$ (green trajectory), $r = 3$ (red trajectory) and initial condition at the location of the base state for $\lambda = \lambda_-$. Figures 2 (c), (f) are obtained using parameter values: $T = 500$, $s = -4$, $A = 1.5$ $\lambda_- = -0.1$, $\Delta_\lambda = -0.4$, $r = 1$ (green trajectory), $r = 3$ (red trajectory) and initial condition at the location of the base state for $\lambda = \lambda_-$. Figure 3 was obtained using a value of $T$ such that $a(0)$ is no larger than $10^{-4}$ and other parameter values: $s = 4$, $A = 3.2$, $\lambda_- = -0.5$, $r = 0.5$ (blue trajectory), $r = 0.9$ (orange trajectory), $r = 1.7$ (purple trajectory) and initial condition at the location of the base state for $\lambda = \lambda_-$.

**Plant-herbivore model**

The time evolution of plant, $P$, and herbivore, $H$, biomass densities [g/m$^2$] can be modelled as two coupled ordinary differential equations (Scheffer et al., 2008; O'Keeffe and Wieczorek, 2020).

$$\frac{dP}{dt} = \rho(rt)P - CP^2 - Hg(P), \tag{4}$$

$$\frac{dH}{dt} = H\left(Ee^{-bP}g(P) - m(rt)\right), \tag{5}$$

with a non-monotone functional response

$$g(P) = c_{\max}\frac{P^2}{P^2 + a^2}e^{-b_cP}. \tag{6}$$

Following the approach of O'Keeffe and Wieczorek (2020), we fix six of the eight parameters (see Table 2.1 Ref. (O'Keeffe and Wieczorek, 2020)) and allow the plant growth rate $\rho(rt)$ [1/day] and herbivore mortality rate $m(rt)$ [1/day] to vary over time subject to environmental conditions. In this system, the external forcing is given by

$$\lambda(rt) = \begin{pmatrix} \rho(rt) \\ m(rt) \end{pmatrix} = \begin{pmatrix} \rho_- + \Delta_\rho a(rt) \\ m_- + \Delta_m a(rt) \end{pmatrix}, \tag{7}$$

where $a(rt)$ takes the form of a ramp or a return profile , and $\rho(rt)$ and $m(rt)$ are varied proportionally to each other according to

$$m(rt) = g(\rho(rt) - \rho_-) + m_-, \tag{8}$$

where $g = 1/30$. . The forcing profile of the environmental conditions, $e(rt)$, corresponding to a normalised shift in $\rho$ and $m$, is given by

$$e(rt) = \Delta_e a(rt). \tag{9}$$

The peak change in environmental conditions, used in Figure 5 (a), is given by

$$\Delta_e = \frac{\Delta_\rho}{\Delta_{\rho,\text{Fold}}} = \frac{\Delta_m}{\Delta_{m,\text{Fold}}}, \tag{10}$$

where $\Delta_{\rho,\text{Fold}} = 0.1966$ and $\Delta_{m,\text{Fold}} = 0.00655$.

Figures 4(a)-(c) are obtained using parameter values: $T = 200$, $\rho_- = 0.5$, $m_- = 0.125$, $\Delta_\rho = 0.1$, $\Delta_m = 0.0033$ ($\Delta_e = 0.51$), $r = 0.03$ (green trajectory), $r \approx 0.0453$ (blue trajectory), $r = 0.1$ (red trajectory) and initial condition at the location of the base state for $\rho = \rho_-$ and $m = m_-$.

Figure 5 (a) is obtained using a value of $T$ such that $a(0)$ is no larger than $10^{-5}$ and initial condition at the location of the base state for $\rho = \rho_-$ and $m = m_-$ such that it is equal to 1 at the Fold bifurcation.

## AMOC model

Wood et al. (2019) proposed a five-box model to model the global oceanic current. The model is driven by salinity fluxes [psu], $S_i$, in the main ocean waters: the North Atlantic ($N$), Tropical Atlantic ($T$), Indo-Pacific ($IP$), Southern Ocean ($S$) and Bottom waters ($B$). The flow strength, $q$ [Sv], of the Atlantic Meridional Overturning Circulation (AMOC) is subsequently determined by a (fixed) temperature gradient, $\Delta T$, and variable salinity gradient, $\Delta S = S_N - S_S$, between the North Atlantic and Southern Ocean boxes:

$$q = \frac{\lambda(\alpha \Delta T + \beta \Delta S)}{1 + \lambda \alpha \mu}. \tag{11}$$

Alkhayuon et al. (2019) empirically highlighted that the salinity of the Southern Ocean and Bottom waters, vary much slower than the other boxes. Therefore, assuming these salinities are fixed and the salinity in the Indo-Pacific, $S_{IP}$, can be determined from a conservation of salinity, the original model reduces to the following two dimensional model:

$$V_N \frac{dS_N}{dt} = q(S_T - S_N) + K_N(S_T - S_N) - F_N(rt)S_0, \tag{12}$$

$$V_T \frac{dS_T}{dt} = q[\gamma S_S + (1-\gamma)S_{IP} - S_T] + K_S(S_S - S_T) + K_N(S_N - S_T) - F_T(rt)S_0, \tag{13}$$

for $q \geq 0$ and

$$V_N \frac{dS_N}{dt} = |q|(S_B - S_N) + K_N(S_T - S_N) - F_N(rt)S_0, \tag{14}$$

$$V_T \frac{dS_T}{dt} = |q|(S_N - S_T) + K_S(S_S - S_T) + K_N(S_N - S_T) - F_T(rt)S_0, \tag{15}$$

for $q < 0$. For more details about the parameters and their values we refer the reader to Tables 3 and 4 in Alkhayuon et al. (2019).

In this study we are interested in the surface freshwater fluxes $F_i$ [Sv], $i$ in $\{N, T, IP, S\}$ as input parameters for the system. Following the approach of Alkhayuon et al. (2019); Wood et al. (2019) these fluxes are defined as linear functions of a hosing parameter $H$ [Sv] such that: the total flux is 0 for all $H$, and $H = 0$ corresponds to the baseline values of $F_i$ (Table 3

Ref.(Alkhayuon et al., 2019)). In this system, the external forcing is given by:

$$\lambda(rt) = \begin{pmatrix} F_N(rt) \\ F_T(rt) \\ F_{IP}(rt) \\ F_S(rt) \end{pmatrix} = \begin{pmatrix} 0.486 + 0.1311\, H(rt) \\ -0.997 + 0.6961\, H(rt) \\ -0.754 - 0.5646\, H(rt) \\ 1.265 - 0.2626\, H(rt) \end{pmatrix}, \tag{16}$$

where

$$H(rt) = \Delta_H a(rt),$$

and $a(rt)$ takes the form of a ramp or a return profile. Figures 4(d)-(f) are obtained using parameter values: $T = 1000$, $\Delta_H = 0.365$, $r = 0.005$ (green trajectory), $r \approx 0.01645$ (blue trajectory), $r = 0.017$ (red trajectory) and initial condition at the location of the base state (the AMOC-on state) for $H = 0$. Figure 5 (b) was obtained using a value of $T$ such that $a(0)$ is no larger than $10^{-5}$ and initial condition at the location of the base state for $H = 0$.

**The power grid model**

We use a 3-bus power system model from (Dobson et al., 1988; Dobson and Chiang, 1989) to represent the dynamics that can be observed on a power grid. Two generators supply power to a P-Q load in parallel with a capacitor and induction motor (Revel et al., 2006). The model consists of four differential equations for the generator phase angle, $\delta_m$, and angular velocity, $\omega_m$, and the phase angle, $\delta$ and magnitude, $V$, of the load voltage (Ajjarapu and Lee, 1992):

$$\dot{\delta}_m = \omega_m,$$
$$\dot{\omega}_m = \frac{1}{M}\left(-d_m\omega_m + P_m - P_e\right),$$
$$\dot{\delta} = \frac{1}{K_{q\omega}}\left(-K_{qv}V - K_{qv^2}V^2 + Q_l - Q_0 - Q_1(rt)\right),$$
$$\dot{V} = \frac{1}{TK_{q\omega}K_{pv}}\left(K_{p\omega}K_{qv^2}V^2 + (K_{p\omega}K_{qv} - K_{q\omega}K_{pv})V\right.$$
$$\left. + K_{p\omega}(Q_0 + Q_1(rt) - Q_l) - K_{q\omega}(P_0 + P_1 - P_l)\right). \tag{17}$$

where $P_e, P_l$ and $Q_l$ are given by

$$P_l = -V_0 V T_0 \sin(\delta + \theta_0) - V_m V Y_m \sin(\delta - \delta_m + \theta_m) + (Y_0 \sin(\theta_0) + Y_m sin(\theta_m))V^2,$$
$$Q_l = V_0 V Y_0 cos(\delta + \theta_0) + V_m V Y_m \cos(\delta - \delta_m + \theta_m) - (Y_0 \cos(\theta_0) + Y_m \cos(\theta_m))V^2)$$
$$P_e = -V_m V Y_m \sin(\delta - \delta_m \theta_m) - V_m^2 Y_m \sin(\theta_m). \tag{18}$$

In this system, the external forcing is in the form of the reactive power demand (referred to power demand in the main text) of the load, $Q_1(rt)$, and is given by:

$$\lambda(rt) = Q_1(rt) = Q_{1_-} + \Delta_{Q_1} a(rt), \tag{19}$$

where $a(rt)$ takes the form of a ramp or a return profile. Figures 6 (a), (b) are obtained using parameter values: $T = 0.15$, $Q_{1_-} = 5$, $\Delta Q = 5$, $r = 50$ (blue trajectory), $r = 130$ (orange trajectory), $r = 400$ (purple trajectory) and initial condition at the location of the base state at $Q_1 = Q_{1_-}$. Figure 6 (c) is obtained using a value of $T$ such that $a(0)$ is no larger than $10^{-2}$ and parameter value: $Q_{1_-} = 5$ and initial condition at the location of the base state at $Q_1 = Q_{1_-}$

For a full description of the other parameters and the values used in this study, we refer the reader to (Budd and Wilson, 2002). However, we choose to set $P_1 = 5$, the real power demand of the load, such that when $Q_1(rt)$ is varied, according to Equation (19), the system can only cross a fold bifurcation and does not encounter a Hopf bifurcation as observed for $P_1 = 0$ (Wang et al., 1994).

*Code availability.* The codes used to conduct simulations and generate figures are available via the GitHub repository (Ritchie et al., 2022).

*Video supplement.* supplementary videos are available via the GitHub repository (Ritchie et al., 2022).

*Competing interests.* There are no competing interests

*Acknowledgements.* P.R. and P.C. research was funded by the European Research Council 'Emergent Constraints on Climate-Land feedbacks in the Earth System (ECCLES)' project, grant agreement number 742472. H.A. and S.W. research was funded by Enterprise Ireland grant no. 20190771.

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
