# Peer review of "Rate-induced tipping in natural and human systems"

_EGUsphere, 2022_

## Author Comment (AC1)

**Rate-induced tipping in natural and human systems**
**Reviewer 1 Responses**

We are grateful for the constructive reviewer comments received on our manuscript. These comments are repeated below in italic type. Our responses are coloured blue and given in normal type. New text copied from our revised manuscript is presented in quotations.

**Response to Reviewer 1**

The authors submitted a great, concise review of the phenomenon of rate-induced tipping and showcase it nicely using conceptual models from ecology, climate, and powergrids. I very much enjoyed reading this. The authors accomplished to explain rate-induced tipping in a very accessible manner while being technically fully accurate and still giving sufficient details to allow for an easy reproduction of the examples. The authors might want to take into account some of the very minor comments below; but the manuscript could also be accepted as is I think.

Some minor comments / questions / suggestions (and most are really very minor):

- l6: "an instability that occurs when external forcing varies across some critical rate" - to me it seems this could be misunderstood, maybe "varies faster than some critical rate"
  Agreed, we will make the change as suggested.

- l11: I wouldn't necessarily say that the "changes" are referred to as "tipping points", the latter are rather the points (in forcing or time) at which such changes occur?
  Yes, indeed this was an oversight on our part, we will make the following revision "The points in time, or the level of forcing, at which such changes occur are commonly referred to as *bifurcation-induced tipping points*."

- l19: I think that the seminal paper by Stocker Schmittner (https://www.nature.com/articles/42224) should be cited here as well - to my knowledge this is the first paper describing rate-induced tipping effects, at least in the climate context
  Thank you we will add this reference in the suggested location

- l52: replace "vanishing" by "sufficiently small"? Also in the next line, "instantaneously" to "fast enough"?
  Changes will be made as suggested

- l76: istability → instability
  Thank you, we will correct this

- Fig.3: is the axis of (c) logarithmic?
  Yes it is indeed logarithmic, we will add the following to the caption "... note the logarithmic scale for the rate parameter."

- l109: could this sentence be simplified?
  We will simplify this sentence by splitting it into two, writing "However, the added possibility of R-tipping, combined with the symmetric return forcing (see Methods for further details), means that *multiple critical rates* can arise for return forcing profiles with the same peak level. These multiple rates emerge due to the competition between the slow approach rates required to avoid R-tipping and the fast return (and approach) rates required for safe overshoots."

- l125: the system has too much intertia?
  The system has too much inertia for the forcing but agree this can be phrased better and so will make the following revision "Then, for even higher rates (the green region to the right of the upper part of the vertical Fold line), both B-tipping and R-tipping upon return can be prevented since the **system processes are not sufficiently fast** to react to a short forcing impulse."

- l164: cite some more papers, including some of the older ones, on AMOC collapse here as well?
  We will make revisions to the text that will include additional older references.

- Fig.4: show additional panel similar to the one in Fig.5c here as well?
  We will create an additional Figure showing the tipping diagrams for the plant-herbivore and AMOC models and provide accompanying text. We believe that adding two panels to the current Figure 4, which already contains 6 panels, will contain too much information for a single figure.

Niklas Boers

---

## Author Comment (AC2)

**Rate-induced tipping in natural and human systems**
**Reviewer 2 Responses**

We are grateful for the constructive reviewer comments received on our manuscript. These comments are repeated below in italic type. Our responses are coloured blue and given in normal type. New text copied from our revised manuscript is presented in quotations.

**Response to Reviewer 2**

*The article "Rate-induced tipping in natural and human systems" explains the phenomenon that occurs when an accelerating parameter change drives a system's state across an unstable equilibrium into another dynamical regime. The authors demonstrate this using two idealized (artificial) systems, and three simple models supposed to represent ecological, climatological and technological systems, each based on a few coupled ordinary differential equations.*

*I think that the content is relevant and within the scope of the journal. The article also does a relatively good job at explaining rate-induced tipping. On the other hand, I also have one very major point of concern:*

*To put it bluntly: Do we really need another article that demonstrates what rate-induced tipping is? What makes this one different? Scanning the authors reference list, I find several previous articles with a similar aim and scope, for example Alkhayuon and Ashwin 2018, Ashwin et al. 2012, Ashwin and Newman 2021, Scheffer et al. 2008, Wieczorek et al. 2011, Wieczorek et al. 2021.*

*In its current state, the article has a flavor of an "understandable science" communication or teaching essay for academics. I sympathize a lot with such educational articles, but I am also a bit skeptical if a scientific journal is the best place for such a piece, if there is not also some new content. I hence believe that the aim of the article should go beyond a mere demonstration of the phenomenon.*

*I also believe that there are elements in the article already that can fulfil this requirement, but that could be worked out more explicitly. The authors should therefore reconsider what is the main objective and the type of this article, and make this clearer.*

*I can imagine three options to achieve this:*

1. *Extend the content toward a comprehensive review article. The authors already call it a "review", but in my view it is too superficial and selective to deserve that name. The title and abstract promise an extremely broad scope, but what we get is an explanation of the phenomenon with selective examples from very simple models. There is hardly any discussion of R-tipping in other systems and models. From a review I would expect a more comprehensive coverage of phenomena and the state of research.*

2. *Focus on arguments why R-tipping has been underrated. The authors say that rate-induced tipping is "arguably of even greater relevance" than tipping at a critical level. But I wonder why that should be the case in practice. If the authors want to focus on such a particular statement, like in a "perspective" article, then they should present arguments that support their statement.*

3. *The study could also be a normal research article. However, there have been many studies about rate-induced tipping already, as can be seen from the reference list. So the question arises what is new. The phenomenon shown in Fig 3 and 4, that merely increasing the rate of forcing can cause a complex sequence of regimes, looks interesting enough to me, and could possibly be the main result in that case, but it should be checked if/how it has been captured by previous studies. The authors could demonstrate this behavior in some more detail, adjust the abstract and conclusions accordingly, also including examples and a discussion of generality and relevance of the phenomenon. What properties do dynamical systems need to have in order to see such a complex sequence? And how likely would it be to see something like that in complex models and the real world?*

*Option 1 or 3 make the most sense to me, but I don't wish to push the authors (or journal) into one particular direction, as long as the revised article is convincing (i.e. unique, focused, and with well-defined*

and sufficiently comprehensive content).

We also prefer Option 1, although we are deliberately stopping short of a comprehensive review for dynamical system experts. Our paper was motivated by a recognition that there is very little understanding of rate-induced tipping beyond a relatively small group of dynamical system experts (and we suspect that the reviewer may be in that latter select group). Two of the authors of this paper (Cox and Ritchie) saw clear evidence of this when drafting text on climate tipping points for the Intergovernmental Panel on Climate Change $6^{th}$ Assessment Report. Whereas it was possible to get text into the IPCC report related to bifurcation and noise-induced tipping, the text that we had drafted on rate-induced tipping was removed due to a lack of understanding of the phenomenon, or of its relevance to a strongly-forced system such as the climate. We therefore wrote this paper as an easy introduction to the large constituency of readers (especially of a journal like *Earth System Dynamics*) who know rather little about rate-induced tipping. In response to this comment from the reviewer, we have however significantly added to the relevant literature that we cite.

I have two related points of major importance:

1. Choice of methods

   The examples the authors show are very general and simple, and rather repetitions of conceptual models instead of independent / emergent phenomena from process-based models or observations. I would like to read some more statements about their relevance: Has rate-induced tipping been observed in ecosystems, climate or power grids, in a way that gives credibility to the applied models?

   The plant-herbivore model is particularly vague. What kinds of species and ecosystems should I think of? Horses in a steppe? Sea urchins in marine kelp forests? Slugs on the salad in my garden? What is the observed behaviour that the model is supposed to represent? I guess there is information in the cited literature, but at least a few lines would help here.

   The model is proposed by Scheffer *et. al.* (2008) to conceptually study critical transitions in ecosystems in general. In Section 4 we will write "The model has been proposed to conceptually study tipping points in bistable ecosystems. Examples of such systems can be: the dominance shift between submerged macrophytes and phytoplankton (Scheffer *et. al.* 1993), coral reefs and macro-algae (Hughes 1994), or the transition of kelp forests into sea urchin barrens that are dominated by crustose coralline algae (Steneck *et. al.* 2002)."

   The climate example looks more convincing to me because it is based on physical processes and has variables with a specific meaning. However, a bridge to more complex models is missing, where R-tipping has long been studied as well.

   We will add the following text on the AMOC to bridge the gap to complex models showing R-tipping, " Specifically, Stocker and Schmittner (1997) and Lohmann and Ditlevsen (2021) show in coupled climate models that the AMOC can collapse under fast rates of change in either $CO_2$ emissions or freshwater forcing. Additionally, rate-induced tipping of the AMOC has been observed in a global oceanic box model (Alkhayuon et al., 2019)."

   And how well does the power grid model simulate behavior in actual (inter-)national power grids? Is there evidence for R-tipping in these grids?

   The near blackout following the conclusion of the Euro semi final is arguably an example of a near R-tipping. The national grid were prepared for the increased power demand but had failed to anticipate the exact timing. Thus when the surge did occur it happened too quickly for the controllers to adapt and only narrowly avoided the blackout. We will add the following paragraph explaining this, "The latter example in particular was arguably the result of rate-induced tipping effects. The power demand on the network, following the conclusion of the football match, was expected to be high. Hence, the national grid took measures to ensure the network would be able to cope with the high power demand. However, the national grid failed to envisage the match going to extra time and penalties. Thus, the rapid increase in power demand, following the eventual conclusion of the match, gave controllers insufficient time to react."

In general, the linkages between the conceptual models and the real world should be discussed and substantiated more.

2. Title

   The title suggests an extremely broad scope – "natural and human" is virtually everything. It could be OK for a very comprehensive review, but it mislead me a bit in case of the current draft. I suggest to make it more precise to better match the content. "Natural systems" here refers to a brief example from climate research and one from ecology. "Human systems" again is quite vague; I first expected something like societal networks here. A better title for the current article might be "rate-induced tipping in climate, ecological and technological systems"? Of course, the new title should reflect what choice the authors make regarding the aim and scope as discussed above.
   While the proposed revised title is fine, we do not think it is a significant improvement on the current title (which was chosen to express the ubiquity of rate-induced tipping in human and natural systems). Also, we prefer to retain the current title in the interests of continuity between the pre-print and the final published paper.

   The selection of content the authors want to focus on could be better justified. What are the criteria? Why exactly grazing, ocean circulation, and power grids? The Authors should both limit the scope and extend the content substantially in order to have comprehensive content within the scope.
   We chose these case-studies because they provide distinct examples of rate-induced tipping in important human and natural systems, and are based on dynamical system models of varying complexity. We will make this rationale clearer in our revision.

List of more minor points:

- Abstract: "hot topic" is arguably somewhat informal.
  Agreed, we will change this to "... a topic of a heated scientific debate..."

- Can it actually be distinguished properly what is tipping at a critical level versus a critical rate? The control parameter in a model could represent a flux (like freshwater input per year into the North Atlantic). I suppose that the unit alone cannot be essential for the difference, which should rather be the mathematical structure of the problem. It becomes clear later in the paper that this is indeed the case, but the notion of "rate" in the beginning can be a bit confusing.
  We will add an additional paragraph to the introduction specifying how we quantify critical rates in a uniform way, by writing
  "Characterising rate-induced tipping raises the issue of defining *critical rates* of change in external forcing. On the one hand, different external forcings will have different physical units and be different, often nonlinear functions of time. On the other hand, we would like to quantify critical rates of change in a uniform way, that is independent of the physical units and the temporal shape of the forcing. Therefore, we introduce a *rate parameter $r$* in units inverse second (or day, year, etc.), denote the external forcing as $\Lambda(rt)$, and work with $r$ as the main input parameter. Most importantly, we define a *critical rate* as a special value of $r$ at which rate-induced tipping occurs, while all the other parameters of $\Lambda(rt)$ remain fixed. To avoid confusion, we note that $d\Lambda/dt = r\,d\Lambda(u)/du$ has units of $\Lambda$ per second, depends on $r$ as well as on the shape of $\Lambda(u)$, and may itself be a function of time (For example, if $\Lambda$ is a rate or acceleration of some sort, $r$ will quantify the rate of change of this rate or acceleration, respectively.). For example, in the case of a linear ramp, $\Lambda(rt) = \lambda rt$, we have a constant in time $d\Lambda/dt = r\lambda$. However, in the case of a nonlinear shift, $\Lambda(rt) = \lambda\tanh(rt)$, we have a time-varying $d\Lambda/dt = r\lambda\operatorname{sech}^2(rt)$ with a maximum $r\lambda$."

- I like Fig. 1 in principle. One could add an arrow to indicate movement of the potential landscape to the left. What is a bit unintuitive: It seems that a critical rate alone is still not enough, but the movement of the potential has to be large enough as well (if it moves infinitely fast but the ball stays close to the minimum, nothing happens). In the text, it reads like a critical rate alone is sufficient. Probably this is also the difference to a "B-tipping" where the control parameter represents a rate of change in physical units?

We will add the arrow and label to Fig. 1 as suggested as well as provide more labels as suggested by other reviewers. In our description of Fig. 1 we do already refer to the base state as being threshold unstable and that this is a sufficient for the occurence of rate-induced tipping, writing, " If the threshold moves past the initial position of the well for a new forcing level, as shown in panel (b), the base state is said to be threshold unstable on varying the forcing ... One can prove that, in general, threshold instability is sufficient for the occurrence of rate-induced tipping (Kiers and Jones, 2020; Wieczorek et al., 2021)".

- Something that I find confusing about Fig. 1 is: I have to assume that the ball has no mass (in the sense that I don't need energy to move the potential and/or lift the ball to the hill)? But it does have inertia (otherwise I could not shift the potential left or right)? And: If it has inertia, it would oscillate around the minimum, unless there is large friction. But if there is large friction, how can I pull away the potential? I guess it is hard to find a physical model that is a better analogy, but at least the essentials and limitations of the analogy should be mentioned.
  Although imperfect, the analogy of a ball in a well is frequently used to represent bifurcation-induced tipping. For comparability and reader engagement, we therefore chose to explain rate-induced tipping using a similar analogy. We will however acknowledge the limitations of such analogies, by writing, "We note that this example is for illustrative purposes since not all dynamical systems can be characterised by a stability landscape Zhou et al. (2012)".

- Line 29: what is a "forced system"? One with boundary conditions, or one where boundary conditions change over time, or even where they accelerate? It seems to me that the latter is needed for rate-induced tipping, but acceleration is not mentioned anywhere. In general, "forcing" is used a lot in the article but not well-defined in the beginning (though I got the idea later on that forcing is the control parameter's value while "forced system" implies it's changing over time?).
  We refer to a forced system as a system that is subjected to external disturbances through boundary conditions. These external forcings can have different physical units, can be constant or vary over time in different manners, which is our focus. We will add the following paragraph to the beginning of the introduction:
  "In this paper, we consider instabilities in open nonlinear systems. Such systems are subjected to external disturbances through boundary conditions, which we refer to as *external forcings*. Different forcings will have different physical units. Furthermore, they may be constant or vary over time in different manners, ranging from a linear ramp to complicated nonlinear waveforms. Our focus is on time-varying external forcings that are nonlinear, can be non-monotone, but always decay to a constant."

- Fig. 2: It could make this figure more understandable by showing how the forcing (and state) change in time. Also, for the arguments in the caption to work (e.g.: avoid B-tipping and then cause R-tipping in c), the particular shape of the black curves is important. But these curves are different from typical "saddle-node" bifurcation curves shown in the references. In particular, stable and unstable branches are tilted in the plotted space, and always very close together. So I wonder how generic the "return tipping" is? It looks like a much more special behavior than B or R-tipping in general.
  Thank you for the suggestion, we will include the suggested time series for the external forcing profiles above each panel. Yes, the curves are different from typical "addle-node" examples, which we already acknowledge by writing, "Figure 2 introduces a subtle but crucial difference to previous examples that have considered B-tipping (Lenton et al., 2008; Scheffer et al., 2009; Ritchie et al., 2021) and that is to apply a tilt to the bifurcation structure (O'Keeffe and Wieczorek, 2020, Sec.7." The tilted saddle-node example is arguably the simplest setting for return tipping, but will be more common in higher dimensional systems.

- What are the methods used to plot Fig. 2?
  We will provide a small section in the Methods explaining the conceptual model including specifying the ODE used.

- Fig. 2b and c: I don't understand why the state would suddenly drop to 0 instantly after crossing the dotted line. If it has inertia (as is needed for the tipping to occur), it would not care, but continue on

a curved continuous line.

Yes, this would indeed be the case if the forcing was continually increasing. However, for this example the forcing decays to a constant. We will include time series of the external forcing profiles above each panel to make this clearer.

- Line 80: "then a natural option would be to reverse the external forcing to avoid crossing the critical level." Why? It would suffice to stop the forcing from changing. For example, I don't expect that mankind will reverse greenhouse gas forcing with the same rate as the previous increase, which would be even much more difficult than reaching net zero (and probably unnecessary). Maybe for the power grid this matters, but I don't see the connection between that model and the model used for Fig. 2.

  The reviewer raises an interesting point. However, in the real world we do not know where the critical level is precisely. We may be able to detect that we are approaching a critical level but do not necessarily know the exact location. Therefore, stopping the forcing may not be sufficient as we may have already crossed the critical level without realising. Hence the presumed safest option would be to reverse the forcing as far as possible. To address this we will write, "If a system is thought to be approaching a B-tipping event, then a natural option would be to reverse the external forcing to avoid crossing a largely unknown critical level."

- Fig. 3: a nice complement to Fig 2. But could both Figures show the same example? Unintentional return tipping (like in Fig 2c) does not occur here? It would help a lot to also see the stable and unstable equilibria of this system.

  Fig 3 does show the same example as Figs 2a and 2b, however, does not include Fig 2c. We now make this clearer in the text writing, "The corresponding response of the system, used in Figure 2(a) and (b), subject to these external forcing trajectories is depicted in Figure 3(b).". Including the tipping diagram for Fig 2c as well would be too much. While return tipping is an interesting result and worth raising awareness to the reader it is not the main focus of the study. Note that the equilibria (for the static system) when plotted against time will be different due to the varying rates of forcing, combined with them being shown in Fig 2 we do not believe they are needed here too.

- Fig 3a: black = blue+purple+orange?

  Yes, we had written this in the text but will also write the following in the captions for Figs. 3 and 5: "Ramp forcing profiles are given by a concatenation of the left half of a colour curve and the black dashed curve."

- Line 98: "previous research has shown that..." Isn't that the definition of B-tipping, not a research result?

  Correct, we will delete this part and make the following amendment: "B-tipping occurs if the external forcing crosses the Fold bifurcation without returning."

- Fig 3c: How much does this rely on the particular shape of the function forcing versus time? At least it seems to require symmetry in the ramp up and down phases. This is a very strong and, if you think of real-world examples, restrictive assumption.

  A sech type return forcing, that is indeed symmetric, is the natural first choice return forcing profile to consider. The quantitative picture for the tipping diagram (3c) will change based on the forcing profile considered, however, qualitatively the same regions still exist.

- Line 109-110: I don't really understand the statement about "multiple critical rates" / slow and fast rates. Wouldn't any accelerated ramp up require one specific minimum rate of ramp down (given a certain function shape)? But in the system the authors used to generate Fig. 3 (equations would be nice), the ramp up is always assumed to be symmetric to the ramp down? This leads to the "white-green-red-green" regimes when increasing the overall rate. This behavior is indeed interesting; but how generic is it? How does this system differ from the stability diagram in Fig 2?

  Yes it is true that for one specific rate of ramp up there will be a single minimum rate of ramp down to avoid tipping. However, as the reviewer notes the multiple critical rates arises because of the symmetric nature of the return forcing. For faster approaches, this increases the likelihood of R-tipping but the corresponding faster return rates aid the possibility of safely overshooting. We will make the following

text changes to address these points: "However, the added possibility of R-tipping, combined with the symmetric return forcing (see Methods for further details), means that *multiple critical rates* can arise for return forcing profiles with the same peak level. These multiple rates emerge due to the competition between the slow approach rates required to avoid R-tipping and the fast return (and approach) rates required for safe overshoots." The tipping diagram presented in Fig 3c corresponds to the system shown in Fig 2a & 2b.

- Line 112 and elsewhere: "fixed maximal change" is confusing. Do the authors mean the amplitude of the Forcing pulse? Or the maximum rate of change? And "Fold level" is the bifurcation point of the static system?
  In this instance we do indeed mean a fixed amplitude of the forcing pulse, however, we also want to exclude the monotone ramp forcing profiles and so we will change this and other instances to a "fixed peak change". Yes, fold level is the bifurcation point of the static system this will also be clarified.

- At times, the authors cite rather selectively, e.g. only very recent research, and make the impression that rate-induced tipping phenomena are a rather new field of study. But this is not the case. For instance, as one of the other reviewers points out, rate-induced collapse of the ocean circulation has been a known phenomenon in complex climate models at least since the 1990ies, https://www.nature.com/articles/42224. As stated above, if the paper is supposed to be a review, the reference list appears rather short.
  We will cite more references, including older references on rate-induced tipping in the ocean circulation.

- Line 195-200: unclear to me, could be better explained. There are infinitely many "stable equilibria", called base states. If I shift the phase by 2pi, don't I get the same behaviour again, instead of a different solution? Here it says "see Methods", but I don't find an answer there. Then, despite the infinite number of stable equilibria there are only two "alternative states". Each base state has only one specific alternative transient state? Why are the other stable equilibria not also "alternative states"? The blackout is one alternative state to all the base states, correct?
  We agree this could be phrased better. Shifting the phase by $2\pi$ does indeed result in the same voltage level therefore we will now refer to this as a single base state. This base state has two alternative states one transient corresponding to a phase slip, the other one is a permanent state (i.e. blackout). We will rephrase the relevant text to make these points clearer in the manuscript.

- Line 306: grid, not gird
  Thank you for spotting this typo, this will be corrected.

- I suspect that most readers will not be familiar with at least two out of the three models because these models describe very different phenomena from different scientific fields. A little more background information about these models and ideally a figure about each would be welcome.
  Agreed, we will add further background details on the models used to help the understanding of the reader.

- Video supplement: Videos could be a great supplement. However, I was unable to find the github repo referenced in "Ritchie et al., 2022". Please provide a link that works, and one that works for readers without a github account.
  The videos will become public available at the github repository once the manuscript has been published.

---

## Author Response (AR1)

**Rate-induced tipping in natural and human systems**
**Reviewer Responses**

We are grateful for the constructive reviewer comments received on our manuscript. These comments are repeated below in black text and our responses are coloured blue. New text copied from our revised manuscript is presented in quotations.

**Response to Reviewer 1**

The authors submitted a great, concise review of the phenomenon of rate-induced tipping and showcase it nicely using conceptual models from ecology, climate, and powergrids. I very much enjoyed reading this. The authors accomplished to explain rate-induced tipping in a very accessible manner while being technically fully accurate and still giving sufficient details to allow for an easy reproduction of the examples. The authors might want to take into account some of the very minor comments below; but the manuscript could also be accepted as is I think.

Some minor comments / questions / suggestions (and most are really very minor):

- l6: "an instability that occurs when external forcing varies across some critical rate" - to me it seems this could be misunderstood, maybe "varies faster than some critical rate"
  Agreed, we will make the change as suggested.

- l11: I wouldn't necessarily say that the "changes" are referred to as "tipping points", the latter are rather the points (in forcing or time) at which such changes occur?
  Yes, indeed this was an oversight on our part, we now make the following revision "The points in time, or in the level of forcing, at which such changes occur are commonly referred to as *bifurcation-induced tipping points*."

- l19: I think that the seminal paper by Stocker Schmittner (https://www.nature.com/articles/42224) should be cited here as well - to my knowledge this is the first paper describing rate-induced tipping effects, at least in the climate context
  Thank you we now add this reference in the suggested location

- l52: replace "vanishing" by "sufficiently small"? Also in the next line, "instantaneously" to "fast enough"?
  We have changed these to "sufficiently slow" and "sufficiently fast" respectively.

- l76: istability → instability
  Thank you, this has been corrected.

- Fig.3: is the axis of (c) logarithmic?
  Yes it is indeed logarithmic, we now add the following to the caption "... note the logarithmic scale for the rate parameter."

- l109: could this sentence be simplified?
  We have further elaborated on this sentence, writing "However, the added possibility of R-tipping owing to the tilted bifurcation structure, combined with the symmetric return forcing (see Methods for further details), means that *multiple critical rates* can arise for a fixed peak change in the return forcing profiles; this is illustrated by the S-shaped solid black curve in Figure 3(c). In a symmetric return forcing, multiple critical rates emerge because there is competition between sufficiently slow approach towards the Fold required to avoid R-tipping, and sufficiently fast reversal required for safe overshoots of the Fold."

- l125: the system has too much intertia?
  The system has too much inertia for the forcing but agree this can be phrased better and so we make the following revision "Then, for the same peak change and very large rate parameters (the green region to the right of the upper part of the vertical Fold line), both B-tipping and R-tipping upon return can be prevented again, but for a different reason - the system processes are too slow to react to a fast forcing impulse."

- l164: cite some more papers, including some of the older ones, on AMOC collapse here as well?
  We now refer to the Stocker and Schmittner, 1997 reference in this section of text.

- Fig.4: show additional panel similar to the one in Fig.5c here as well?
  We believe that adding two panels to the current Figure 4, which already contains 6 panels, will contain too much information for a single figure. However, we do agree that the tipping diagrams would be useful for the plant-herbivore and AMOC models and so we have now added them as a new figure (Fig. 5) and refer to it at the end of section 5.

Niklas Boers

**Response to Reviewer 2**

The article "Rate-induced tipping in natural and human systems" explains the phenomenon that occurs when an accelerating parameter change drives a system's state across an unstable equilibrium into another dynamical regime. The authors demonstrate this using two idealized (artificial) systems, and three simple models supposed to represent ecological, climatological and technological systems, each based on a few coupled ordinary differential equations.

I think that the content is relevant and within the scope of the journal. The article also does a relatively good job at explaining rate-induced tipping. On the other hand, I also have one very major point of concern:

To put it bluntly: Do we really need another article that demonstrates what rate-induced tipping is? What makes this one different? Scanning the authors reference list, I find several previous articles with a similar aim and scope, for example Alkhayuon and Ashwin 2018, Ashwin et al. 2012, Ashwin and Newman 2021, Scheffer et al. 2008, Wieczorek et al. 2011, Wieczorek et al. 2021.

In its current state, the article has a flavor of an "understandable science" communication or teaching essay for academics. I sympathize a lot with such educational articles, but I am also a bit skeptical if a scientific journal is the best place for such a piece, if there is not also some new content. I hence believe that the aim of the article should go beyond a mere demonstration of the phenomenon.

I also believe that there are elements in the article already that can fulfil this requirement, but that could be worked out more explicitly. The authors should therefore reconsider what is the main objective and the type of this article, and make this clearer.

I can imagine three options to achieve this:

1. Extend the content toward a comprehensive review article. The authors already call it a "review", but in my view it is too superficial and selective to deserve that name. The title and abstract promise an extremely broad scope, but what we get is an explanation of the phenomenon with selective examples from very simple models. There is hardly any discussion of R-tipping in other systems and models. From a review I would expect a more comprehensive coverage of phenomena and the state of research.

2. Focus on arguments why R-tipping has been underrated. The authors say that rate-induced tipping is "arguably of even greater relevance" than tipping at a critical level. But I wonder why that should be the case in practice. If the authors want to focus on such a particular statement, like in a "perspective" article, then they should present arguments that support their statement.

3. The study could also be a normal research article. However, there have been many studies about rate-induced tipping already, as can be seen from the reference list. So the question arises what is new. The phenomenon shown in Fig 3 and 4, that merely increasing the rate of forcing can cause a complex sequence of regimes, looks interesting enough to me, and could possibly be the main result in that case, but it should be checked if/how it has been captured by previous studies. The authors could demonstrate this behavior in some more detail, adjust the abstract and conclusions accordingly, also including examples and a discussion of generality and relevance of the phenomenon. What properties

do dynamical systems need to have in order to see such a complex sequence? And how likely would it be to see something like that in complex models and the real world?

Option 1 or 3 make the most sense to me, but I don't wish to push the authors (or journal) into one particular direction, as long as the revised article is convincing (i.e. unique, focused, and with well-defined and sufficiently comprehensive content).

We also prefer Option 1, although we are deliberately stopping short of a comprehensive review for dynamical system experts. Our paper was motivated by a recognition that there is very little understanding of rate-induced tipping beyond a relatively small group of dynamical system experts (and we suspect that the reviewer may be in that latter select group). Two of the authors of this paper (Cox and Ritchie) saw clear evidence of this when drafting text on climate tipping points for the Intergovernmental Panel on Climate Change $6^{th}$ Assessment Report. Whereas it was possible to get text into the IPCC report related to bifurcation and noise-induced tipping, the text that we had drafted on rate-induced tipping was removed due to a lack of understanding of the phenomenon, or of its relevance to a strongly-forced system such as the climate. We therefore wrote this paper as an easy introduction to the large constituency of readers (especially of a journal like *Earth System Dynamics*) who know rather little about rate-induced tipping. In response to this comment from the reviewer, we have however significantly added to the relevant literature that we cite.

I have two related points of major importance:

1. Choice of methods

   The examples the authors show are very general and simple, and rather repetitions of conceptual models instead of independent / emergent phenomena from process-based models or observations. I would like to read some more statements about their relevance: Has rate-induced tipping been observed in ecosystems, climate or power grids, in a way that gives credibility to the applied models?

   The plant-herbivore model is particularly vague. What kinds of species and ecosystems should I think of? Horses in a steppe? Sea urchins in marine kelp forests? Slugs on the salad in my garden? What is the observed behaviour that the model is supposed to represent? I guess there is information in the cited literature, but at least a few lines would help here.

   The model is proposed by Scheffer *et. al.* (2008) to conceptually study critical transitions in ecosystems in general. In Section 5 (previously Section 4) we write "The model has been proposed to conceptually study tipping points in bistable ecosystems with a non-monotone functional response. Examples of such systems can be: the dominance shift between submerged macrophytes and phytoplankton (Scheffer *et. al.* 1993), coral reefs and macro-algae (Hughes 1994), or the transition of kelp forests into sea urchin barrens that are dominated by crustose coralline algae (Steneck *et. al.* 2002)."

   The climate example looks more convincing to me because it is based on physical processes and has variables with a specific meaning. However, a bridge to more complex models is missing, where R-tipping has long been studied as well.

   We now add the following text on the AMOC to bridge the gap to complex models showing R-tipping, " Specifically, Stocker and Schmittner (1997) and Lohmann and Ditlevsen (2021) show in coupled climate models that the AMOC can collapse under sufficiently fast rates of change in either $CO_2$ emissions or freshwater hosing. Additionally, R-tipping of the AMOC has been observed in a global oceanic box model (Alkhayuon et al., 2019)."

   And how well does the power grid model simulate behavior in actual (inter-)national power grids? Is there evidence for R-tipping in these grids?

   The near blackout following the conclusion of the Euro semi final is arguably an example of a near R-tipping. The national grid were prepared for the increased power demand but had failed to anticipate the exact timing. Thus when the surge did occur it happened too quickly for the controllers to adapt and only narrowly avoided the blackout. We add the following paragraph explaining this, "The latter example in particular was arguably the result of R-tipping effects. The power demand on the network, following the conclusion of the football match, was expected to be high. Hence, the national grid took

measures to ensure the network would be able to cope with the high power demand. However, the national grid failed to envisage the match going to extra time and penalties. Thus, the rapid increase in power demand, following the eventual conclusion of the match, gave controllers insufficient time to react."

In general, the linkages between the conceptual models and the real world should be discussed and substantiated more.

2. Title

The title suggests an extremely broad scope – "natural and human" is virtually everything. It could be OK for a very comprehensive review, but it mislead me a bit in case of the current draft. I suggest to make it more precise to better match the content. "Natural systems" here refers to a brief example from climate research and one from ecology. "Human systems" again is quite vague; I first expected something like societal networks here. A better title for the current article might be "rate-induced tipping in climate, ecological and technological systems"? Of course, the new title should reflect what choice the authors make regarding the aim and scope as discussed above.
While the proposed revised title is fine, we do not think it is a significant improvement on the current title (which was chosen to express the ubiquity of rate-induced tipping in human and natural systems). Also, we prefer to retain the current title in the interests of continuity between the pre-print and the final published paper.

The selection of content the authors want to focus on could be better justified. What are the criteria? Why exactly grazing, ocean circulation, and power grids? The Authors should both limit the scope and extend the content substantially in order to have comprehensive content within the scope.
We chose these case-studies because they provide distinct examples of rate-induced tipping in important human and natural systems, and are based on dynamical system models of varying complexity. We now make this rationale clearer throughout the manuscript.

List of more minor points:

- Abstract: "hot topic" is arguably somewhat informal.
  Agreed, we have changed this to "... a topic of a heated scientific debate..."

- Can it actually be distinguished properly what is tipping at a critical level versus a critical rate? The control parameter in a model could represent a flux (like freshwater input per year into the North Atlantic). I suppose that the unit alone cannot be essential for the difference, which should rather be the mathematical structure of the problem. It becomes clear later in the paper that this is indeed the case, but the notion of "rate" in the beginning can be a bit confusing.
  We have added a new Section titled "Defining critical rates" to address this comment. In this section we write
  "Let's denote the time-varying external forcing with $\lambda$. The level of the forcing at a time $t$ is simply the value of $\lambda$ at this time $t$. However, defining critical rates of change in external forcing is more subtle. On the one hand, different external forcings will have different physical units and be different, often nonlinear functions of time. On the other hand, we would like to quantify critical rates of change in a uniform way, that is independent of the physical units and the temporal profile of the forcing. Therefore, we introduce a rate parameter $r$ in units inverse second (or day, year, etc.), write the external forcing as $\lambda(rt)$, where $u = rt$ is dimensionless, and work with $r$ as the main input parameter. Most importantly, we define a critical rate as a special value of r at which rate-induced tipping occurs, while the shift magnitude of $\lambda(rt)$ remains fixed."
  Additionally, we are now careful to distinguish "between the rate parameter $r$ and the rate of change of external forcing $d\lambda/dt$". We also note that "if the forcing $\lambda$ itself is a rate of some sort, $\lambda(rt)$ will be the level of this rate at time $t$, referred to as the level of the forcing, and $r$ will quantify the rate of change of this rate, referred to as the rate of change of the forcing."

- I like Fig. 1 in principle. One could add an arrow to indicate movement of the potential landscape to the left. What is a bit unintuitive: It seems that a critical rate alone is still not enough, but the

movement of the potential has to be large enough as well (if it moves infinitely fast but the ball stays close to the minimum, nothing happens). In the text, it reads like a critical rate alone is sufficient. Probably this is also the difference to a "B-tipping" where the control parameter represents a rate of change in physical units?

We have added the arrow and label to Fig. 1 as suggested as well as provide more labels as suggested by other reviewers. In our description of Fig. 1 we do already refer to the base state as being threshold unstable and that this is sufficient for the occurrence of rate-induced tipping, writing, " If the threshold moves past the initial position of the base state for a new forcing level, as shown in panel (b), the base state is said to be threshold unstable on varying the forcing ... The threshold/basin instability condition gives the forcing shift magnitude that enables rate-induced tipping. In general, one can prove that threshold/basin instability is sufficient for the occurrence of rate-induced tipping: There is an external forcing that gives rate-induced tipping if the system is threshold/basin unstable (Kiers and Jones, 2020; Wieczorek et al., 2021)".

- Something that I find confusing about Fig. 1 is: I have to assume that the ball has no mass (in the sense that I don't need energy to move the potential and/or lift the ball to the hill)? But it does have inertia (otherwise I could not shift the potential left or right)? And: If it has inertia, it would oscillate around the minimum, unless there is large friction. But if there is large friction, how can I pull away the potential? I guess it is hard to find a physical model that is a better analogy, but at least the essentials and limitations of the analogy should be mentioned.

Although imperfect, the analogy of a ball in a well is frequently used to represent bifurcation-induced tipping. For comparability and reader engagement, we therefore chose to explain rate-induced tipping using a similar analogy. We now however acknowledge the limitations of such analogies, by adding the following footnote, "We note that this example is for illustrative purposes. In general, the base state can be non-stationary, the system may reside near rather than in the base state, and not all dynamical systems can be characterised by a stability landscape; see for example Zhou et al. (2012)".

- Line 29: what is a "forced system"? One with boundary conditions, or one where boundary conditions change over time, or even where they accelerate? It seems to me that the latter is needed for rate-induced tipping, but acceleration is not mentioned anywhere. In general, "forcing" is used a lot in the article but not well-defined in the beginning (though I got the idea later on that forcing is the control parameter's value while "forced system" implies it's changing over time?).

We refer to a forced system as a system that is subjected to external disturbances through boundary conditions. These external forcings can have different physical units, can be constant or vary over time in different manners, which is our focus. We have added the following paragraph to the beginning of the introduction:

"In this paper, we consider tipping instabilities in nonlinear open systems (Ashwin et al., 2012). By "open" we mean systems that are influenced by changing external conditions which we refer to as *external forcings*. In a mathematical dynamic model of an open system, such external forcings are represented by *time-varying input parameters*."

- Fig. 2: It could make this figure more understandable by showing how the forcing (and state) change in time. Also, for the arguments in the caption to work (e.g.: avoid B-tipping and then cause R-tipping in c), the particular shape of the black curves is important. But these curves are different from typical "saddle-node" bifurcation curves shown in the references. In particular, stable and unstable branches are tilted in the plotted space, and always very close together. So I wonder how generic the "return tipping" is? It looks like a much more special behavior than B or R-tipping in general.

Thank you for the suggestion, we now include the suggested time series for the external forcing profiles above each panel. Yes, the curves are different from typical "saddle-node" examples, which we already acknowledge by writing, "Figure 2 introduces a subtle but crucial difference to previous examples that have considered B-tipping (Lenton et al., 2008; Scheffer et al., 2009; Ritchie et al., 2021) and that is to apply a tilt to the bifurcation structure (O'Keeffe and Wieczorek, 2020, Sec.7.)" The tilted saddle-node example is arguably the simplest setting for return tipping, but will be more common in higher dimensional systems.

- What are the methods used to plot Fig. 2?

We have now added a small section in the Methods explaining the conceptual model, including specifying the ODE used and parameter values for the corresponding figures.

- Fig. 2b and c: I don't understand why the state would suddenly drop to 0 instantly after crossing the dotted line. If it has inertia (as is needed for the tipping to occur), it would not care, but continue on a curved continuous line.
  Yes, this would indeed be the case if the forcing was continually increasing. However, for this example the forcing decays to a constant. Following one of your previous comments we now include the time series of the external forcing profiles above each panel, which removes this confusion.

- Line 80: "then a natural option would be to reverse the external forcing to avoid crossing the critical level." Why? It would suffice to stop the forcing from changing. For example, I don't expect that mankind will reverse greenhouse gas forcing with the same rate as the previous increase, which would be even much more difficult than reaching net zero (and probably unnecessary). Maybe for the power grid this matters, but I don't see the connection between that model and the model used for Fig. 2.
  The reviewer raises an interesting point. However, in the real world we do not know where the critical level is precisely. We may be able to detect that we are approaching a critical level but do not necessarily know the exact location. Therefore, stopping the forcing may not be sufficient as we may have already crossed the critical level without realising. Hence the presumed safest option would be to reverse the forcing as far as possible. To address this we will write, "If a system is thought to be approaching a B-tipping event, then a natural option would be to reverse the external forcing to avoid crossing a largely unknown critical level."

- Fig. 3: a nice complement to Fig 2. But could both Figures show the same example? Unintentional return tipping (like in Fig 2c) does not occur here? It would help a lot to also see the stable and unstable equilibria of this system.
  Fig 3 does show the same example as Figs 2a and 2b, however, does not include Fig 2c. We now explicitly state this when introducing Figure 3, "Figure 3 provides a more in-depth analysis of the tilted saddle-node model considered in Figure 2(a), (b)". Including the tipping diagram for Fig 2c as well would be too much. While return tipping is an interesting result and worth raising awareness to the reader it is not the main focus of the study. Note that the equilibria (for the static system) when plotted against time will be different due to the varying rates of forcing, combined with them being shown in Fig 2 we do not believe they are needed here too.

- Fig 3a: black = blue+purple+orange?
  Yes, we had written this in the text but we also now write the following in the captions for Figs. 3 and 5: "Ramp forcing profiles in (a) are given by a concatenation of the left half of a colour curve and the black dashed curve."

- Line 98: "previous research has shown that..." Isn't that the definition of B-tipping, not a research result?
  Correct, we have deleted this part and made the following amendment: "B-tipping occurs if the external forcing crosses the Fold bifurcation without returning."

- Fig 3c: How much does this rely on the particular shape of the function forcing versus time? At least it seems to require symmetry in the ramp up and down phases. This is a very strong and, if you think of real-world examples, restrictive assumption.
  A sech type return forcing, that is indeed symmetric, is the natural first choice return forcing profile to consider. The quantitative picture for the tipping diagram (3c) will change based on the forcing profile considered, however, qualitatively the same regions still exist. We write in the conclusions section "Since our forcing profiles are very idealised by design, we focus on the qualitative behaviour that can arise for different rates of forcing rather than on quantitative predictions. For quantitative predictions, further research on R-tipping is required in more-realistic higher-complexity models, such as state-of-the-art global circulation models, and with more-realistic forcing profiles."

- Line 109-110: I don't really understand the statement about "multiple critical rates" / slow and fast rates. Wouldn't any accelerated ramp up require one specific minimum rate of ramp down (given a

certain function shape)? But in the system the authors used to generate Fig. 3 (equations would be nice), the ramp up is always assumed to be symmetric to the ramp down? This leads to the "white-green-red-green" regimes when increasing the overall rate. This behavior is indeed interesting; but how generic is it? How does this system differ from the stability diagram in Fig 2?

Yes it is true that for one specific rate of ramp up there will be a single minimum rate of ramp down to avoid tipping. However, as the reviewer notes the multiple critical rates arises because of the symmetric nature of the return forcing. For faster approaches, this increases the likelihood of R-tipping but the corresponding faster return rates aid the possibility of safely overshooting. We now write the following to address these points: "However, the added possibility of R-tipping owing to the tilted bifurcation structure, combined with the symmetric return forcing (see Methods for further details), means that *multiple critical rates* can arise for a fixed peak change in the return forcing profiles; this is illustrated by the S-shaped solid black curve in Figure 3(c). In a symmetric return forcing, multiple critical rates emerge because there is competition between sufficiently slow approach towards the Fold required to avoid R-tipping, and sufficiently fast reversal required for safe overshoots of the Fold." The tipping diagram presented in Fig 3c corresponds to the system shown in Fig 2a & 2b. We have also added a small section to the Methods that details the ODE for this conceptual model.

- Line 112 and elsewhere: "fixed maximal change" is confusing. Do the authors mean the amplitude of the Forcing pulse? Or the maximum rate of change? And "Fold level" is the bifurcation point of the static system?

  In this instance we do indeed mean a fixed amplitude of the forcing pulse, however, we also want to include the monotone ramp forcing profiles and so we will change this and other instances to a "fixed peak change". Yes, fold level is the bifurcation point of the static system this will also be clarified.

- At times, the authors cite rather selectively, e.g. only very recent research, and make the impression that rate-induced tipping phenomena are a rather new field of study. But this is not the case. For instance, as one of the other reviewers points out, rate-induced collapse of the ocean circulation has been a known phenomenon in complex climate models at least since the 1990ies, https://www.nature.com/articles/42224. As stated above, if the paper is supposed to be a review, the reference list appears rather short.

  We have cited additional references on R-tipping including: Stocker and Schmittner, 1997; Arumugam et al., 2020; Pierini and Ghil, 2021; Arnscheidt and Rothman, 2022.

- Line 195-200: unclear to me, could be better explained. There are infinitely many "stable equilibria", called base states. If I shift the phase by 2pi, don't I get the same behaviour again, instead of a different solution? Here it says "see Methods", but I don't find an answer there. Then, despite the infinite number of stable equilibria there are only two "alternative states". Each base state has only one specific alternative transient state? Why are the other stable equilibria not also "alternative states"? The blackout is one alternative state to all the base states, correct?

  We agree this could have been phrased better. Shifting the phase by $2\pi$ does indeed result in the same voltage level therefore we will now refer to this as a single base state. There are two alternative states one transient corresponding to a phase slip, the other one is a permanent state (i.e. blackout). We have now made the following amendments to the text to make these points clearer.

  "Since each of these stable equilibria has the same voltage magnitude, they can be thought of as a single base state of the system. Furthermore, there are two alternative states. An *alternative transient state* is a temporary drop in the voltage magnitude accompanied by a $2\pi$ shift in the phase angle. This state corresponds to a transition between two neighbouring stable equilibria within the base state. An *alternative stable state* is at zero voltage magnitude, and corresponds to electrical blackout."

- Line 306: grid, not gird

  Thank you for spotting this typo, this has been corrected.

- I suspect that most readers will not be familiar with at least two out of the three models because these models describe very different phenomena from different scientific fields. A little more background information about these models and ideally a figure about each would be welcome.

Agreed, we have added further background details on the models used to help the understanding of the reader, as specified by our response to the first main comment.

- Video supplement: Videos could be a great supplement. However, I was unable to find the github repo referenced in "Ritchie et al., 2022". Please provide a link that works, and one that works for readers without a github account.
  The videos will become public available at the github repository once the manuscript has been published.

**Response to Reviewer 3**

This manuscript explains the concepts related to the rate-induced tipping and cites the relevant literature along. As such, it is not really a review, I would rather call it a "pedagogical review", and it is somehow up to the editors to decide if it fit the scope of the present journal. My opinion on the subject is that it does fit.

It is well written and could almost be published as is, but I have some suggestions on the first two figures to help the reader (including me).

For Figure 1, I would be more descriptive of what is what in the Figure. I had trouble understanding the links between panel a and b at first glance. I have a proposal below:

Anyway, the authors could change it differently, as long as it becomes clearer.
We thank the reviewer for the suggestions, we now add the following annotations to panel (b) to aid the understanding of the figure: add a black arrow and label indicating the shift in the stability landscape, add labels for the "initial threshold position", "initial stability landscape", and "new stability landscape".

For figure 2, for each panel, I would add besides the external forcing evolution as a function of time for the two different curves, as is done in Figure 4. For it is crucial not to lose the reader at this crucial point.
Agreed, and we thank you for this suggestion, above each panel we add the time series of the external forcing.

Note that the resolution of the figures is not sufficient for printing (screen is ok).
We will provide figures in the required specification by the publishers when applicable.

---

## Author Response (AR2)

**Rate-induced tipping in natural and human systems**
**Reviewer 2 Responses**

We are grateful for the further constructive reviewer comments received on our manuscript. These comments are repeated below in italic type. Our responses are coloured blue and given in normal type. New text copied from our revised manuscript is presented in quotations.

**Response to Reviewer 2**

*Thanks to the discussion and the reformulation of some parts of the article, the aim and target group is now clearer to me, and makes sense. I think that my previous comments have been addressed to an extent that I find sufficient to agree with the publication.*

*I only have some minor additional suggestions:*

- *Sect 2 and 3: Although the applications and demonstrations come later, the authors might give an example here already (in 1-2 sentences). For example, could the case where lambda is in physical units of a rate, be the freshwater flux into the North Atlantic ("hosing") in kg/year (or Sv)?*
  Yes exactly, we now provide the examples of the freshwater flux into the North Atlantic and population growth rates for the predator-prey system. In Section 3, we write "Furthermore, if the forcing $\lambda$ itself is a physical rate of some sort (e.g. freshwater flux into the North Atlantic, measured in Sverdrups – millions of cubic metres per second, or population growth rate measured in individuals per unit area per year, from examples in Section 5)..."

- *line 110 + footnote 2: I now understand why the authors show a stability diagram that deviates from the normal form in a specific way (to demonstrate "return tipping"), but I don't understand why this shape should be "more realistic". It would indeed be unrealistic if the normal form applied far from the saddle-node, but the particular shape the authors chose for demonstration might be even less realistic. I guess what is realistic always depends on the particular system. So I'd suggest to remove the word "realistic" in this context.*
  We have removed "reflects more realistic bifurcation structures and" from the sentence and also removed the word "real" from the footnote.

- *It took me a little while to understand in Fig. 3c why the tipping depends on the forcing rate as shown; what might help further is to plot the trajectories of forcing and state in the same plane as shown in Fig. 2d+e. This is what I meant with showing stable and unstable equilibria in the previous round. The authors point out in their reply that "the equilibria (for the static system) when plotted against time will be different due to the varying rates of forcing". What I meant was not to put time on any axis, but show trajectories in the state vs forcing space as in Fig. 2.*
  Firstly, we would like to remind the reviewer that Figures 2(a,b,d,e) and Figure 3 correspond to the same conceptual model. Ideally, we would include all representations in the same figure but this would be too much. However, we believe it is more important to include the time series representation of the forcing and system response as this is commonly more familiar to climate and Earth system scientists than the state vs forcing plane (that is already plotted in Fig 2).

- *Fig. 5b: The shape of the boundary between the green and red regime looks strangely non-smooth. Is this a numerical bug / lack of simulations and simulation time, or is it (to some extent) related to some non-smooth model property like taking the absolut value of q in Eq. 14+15?*
  The non-smooth appearance is a result of insufficient resolution, instead there would be many small but smooth 'wiggles'. We now write "Additionally the (black) boundary has small 'wiggles' that appear as non-smooth corners. A similar 'wiggling effect' near a Hopf bifurcation has been observed in O'Keeffe and Wieczorek (2020)."

- *Title, line 186 and elsewhere: To me "human systems" still sounds misleading. I would expect some system describing physiological or social behaviour of humans, not the engineering / technology related model of power grids. The authors may still want to consider rephrasing, e.g., human-made systems or technological systems.*
  Although our specific example of a power grid is technological, it is also a human-created system. In

this sense, technological systems are a subset of human systems. In other human systems (e.g. the economy) we also expect to see rate-induced tipping. We therefore prefer to keep 'human' in the title, as this more clearly conveys the ubiquitous nature of rate-induced tipping.

- Line 258-262: It is still unclear to me why (near) power blackouts like during the 1990 semi-final are an example of R-tipping. I understand that the power demand was higher than expected, and that controllers needed time to catch up with the demand, but where does the critical rate come in here? If I interpret power demand minus supply as the forcing, then there would be one critical threshold, which is rather a B-tipping than R-tipping. Where does the memory of the system state come into play that is needed for R-tipping?
  As the reviewer points out, the fact that the controllers nearly did not have enough time to catch up, suggests by definition that it is a rate-induced problem. In this case, there is no issue with providing enough power to meet the new demand level, it is only the "rate" at which the demand is rising that would cause a blackout. We have added the following text to make this clearer, "Therefore in this case, the limiting factor was not the peak in demand, but instead the rate at which the demand on the network rose."

- Line 266, 268 and elsewhere: Still unclear to me how a "$2\pi$ difference in the phase angle" or "a $2\pi$ shift in the phase angle" can be distinguished / defined at all. If I think of just one wave, a $2\pi$ shift gives the identical wave, i.e., 0 phase shift. Why should this be counted as another equilibrium?
  While the equilibria themselves would look the same, the variables are coupled. So, making the transition from one equilibrium to the next equilibrium (separated by a $2\pi$ phase angle) will result in a temporary drop in voltage. We now make this clearer with the following modification to the text, "An *alternative transient state* is a temporary drop in the voltage magnitude accompanied by a $2\pi$ shift in the phase angle, caused by the coupling within the system"

- Line 296: "slowly enough"?
  We have changed this to "sufficiently slowly". Note that in the same sentence we give the opposite side and mention for "...changes faster than some critical rate".

Additional comments I had not raised in the first round (so I think it's appropriate if the authors decide themselves whether they want to make changes or not):

- line 167: "For small overshoots of the Fold, even greater complexity is possible with the potential of three critical rates and two (red) tipping sub-intervals for a fixed peak change of return forcing." Maybe add a figure, e.g., in a supplement?
  This is already highlighted in the tipping diagram (Figure 3(c)), and we do not believe it to be necessary to show these suggested time profiles as the qualitative behaviour has already been shown for all three scenarios (tracking, points of return, points of no return).

- 167-175: the extreme cases of practically infinitely slow or fast forcing discussed here are not shown in Fig. 3c? If the system behaviour changes beyond the range shown, it could be insightful to extend the Figure.
  There is no major change in the behaviour beyond the range shown. In fact, as we explain at the end of page 6, the black dashed line separating tracking from tipping regions asymptotes to the Fold for small $r$ and to the boundary of basin-instability for large $r$. We also already mention on page 8 that the return profile asymptotes to the Fold for small $r$ and for large $r$ tipping is prevented because the "system processes are too slow to react...".

I hope that my comments were somehow helpful to improve the article and compliment the authors for their contribution.
We thank the reviewer again for their helpful and constructive comments.